# Natural killer cells modulate motor neuron-immune cell cross talk in models of Amyotrophic Lateral Sclerosis

Stefano Garofalo[1]✉, Germana Cocozza[2], Alessandra Porzia[3], Maurizio Inghilleri [4], Marcello Raspa[5], Ferdinando Scavizzi [5], Eleonora Aronica[6], Giovanni Bernardini [7], Ling Peng [8], Richard M. Ransohoff[9], Angela Santoni[2,7] & Cristina Limatola [2,10]✉

In amyotrophic lateral sclerosis (ALS), immune cells and glia contribute to motor neuron (MN) degeneration. We report the presence of NK cells in post-mortem ALS motor cortex and spinal cord tissues, and the expression of NKG2D ligands on MNs. Using a mouse model of familial-ALS, hSOD1$^{G93A}$, we demonstrate NK cell accumulation in the motor cortex and spinal cord, with an early CCL2-dependent peak. NK cell depletion reduces the pace of MN degeneration, delays motor impairment and increases survival. This is confirmed in another ALS mouse model, TDP43$^{A315T}$. NK cells are neurotoxic to hSOD1$^{G93A}$ MNs which express NKG2D ligands, while IFNγ produced by NK cells instructs microglia toward an inflammatory phenotype, and impairs FOXP3$^+$/Treg cell infiltration in the spinal cord of hSOD1$^{G93A}$ mice. Together, these data suggest a role of NK cells in determining the onset and progression of MN degeneration in ALS, and in modulating Treg recruitment and microglia phenotype.

[1] Department of Physiology and Pharmacology, Sapienza University of Rome, Rome, Italy. [2] IRCCS Neuromed, Pozzilli, Italy. [3] Department of Experimental Medicine, Sapienza University of Rome, Rome, Italy. [4] Department of Human Neuroscience, Sapienza University, Rome, Italy. [5] EMMA CNR, Monterotondo, Italy. [6] Amsterdam UMC, University of Amsterdam, department of (Neuro)Pathology, Amsterdam Neuroscience, Meibergdreef 9, Amsterdam, The Netherlands. [7] Department of Molecular Medicine, laboratory affiliated to Istituto Pasteur Italia, Sapienza University of Rome, Rome, Italy. [8] Aix-Marseille Université, CNRS, Centre Interdisciplinaire de Nanoscience de Marseille, UMR 7325, «Equipe Labellisée Ligue Contre le Cancer», Marseille, France. [9] Third Rock Ventures, Boston, MA, USA. [10] Department of Physiology and Pharmacology, Sapienza University, Laboratory affiliated to Istituto Pasteur Italia, Rome, Italy. ✉email: stefano.garofalo@uniroma1.it; cristina.limatola@uniroma1.it

Amyotrophic Lateral Sclerosis (ALS) is a fatal neurodegenerative disease characterized by the progressive degeneration of motor neurons (MN)[1]. In a subgroup of familial ALS (fALS), mutant superoxide dismutase 1 (SOD1) is expressed. In this form of ALS and in the corresponding mouse models, microglia acquire an inflammatory phenotype, affecting MN death[2,3], and myeloid cells expressing mutated SOD1 promote neurotoxicity[4,5]. As in other CNS disorders, microglia exert protective or detrimental functions in different disease phases, depending on cell-autonomous features and local microenvironmental cues[6–8]. As one example, the expression level of mutant SOD1 in microglia correlates with the late phase of disease[5], while microglial phenotypes in hSOD1[G93A] mice are modulated by infiltrating Treg cells, through cytokine production[4,9]. Moreover, Treg number and FOXP3 expression are reduced in rapidly progressing ALS patients, and it has been shown that the passive transfer of Tregs prolongs survival, suppressing neuroinflammation[10].

NK cells regulate both the adaptive and innate immune responses, and CNS infiltrating NK cells modulate neuroinflammation in neurodegenerative diseases[11–14], and affect microglial phenotype in cerebral tumors[15,16]. In addition, it was recently described that in pathological conditions, damaged neurons express NKG2D ligands and selectively succumb to NK cell-mediated degeneration[17]. Previous longitudinal cohort studies of peripheral blood from ALS patients revealed an increased number of NK cells in patients[18]; additionally, end-stage hSOD1[G93A] mice showed high NK cell frequency in the spinal cord[19].

Here, we investigate the functional role of NK cells in disease onset and progression in the hSOD1[G93A] mice. We found that NK cells directly kill spinal cord motor neurons in a NKG2D-dependent manner. In addition, we characterized NK cell cross talk with both resident microglia and infiltrating Treg cells in the hSOD1[G93A] mice. We propose that these communication mechanisms orchestrate a vicious cycle of inflammatory signals contributing to onset of motor impairment but not disease progression. Given NK cell presence in postmortem sALS motor cortex and spinal cord, and the expression of NKG2D ligands on postmortem sALS motor neurons, these data suggest a key role of NK cells in ALS onset, highlighting the importance of immune cells in motor neuron loss.

## Results

**NK cell analyses in ALS patients.** To determine whether NK cells were altered in ALS patients (Tables 1, 2), we analysed the CNS tissues and blood of sporadic ALS patients (sALS) for NK cell presence and for frequency of NK cell subtypes. NKp46[+] cells were detected in sALS spinal cord and cerebral motor cortex postmortem tissues while they were not observed in the spinal cord or motor cortex of controls (patients listed in Table 1, Fig. 1a). In addition, peripheral blood from sALS patients showed a reduction of circulating CD56[+]/CD3[−] cells, both in the CD56[high] and CD56[low] cell populations, in comparison with controls (patients listed in Table 2, Fig. 1b).

**NK cells infiltrate motor cortex and spinal cord of hSOD1[G93A] mice.** We then investigated the time dependency of NK cell infiltration in the CNS of hSOD1[G93A] mice. Data in Fig. 2a show that NK cells were present in the spinal cord and motor cortex of hSOD1[G93A] mice, with a peak at the early phase (10–13 weeks), and a decline during the period of motor impairment (16–19 weeks). NKp46[+] cells were absent in these same CNS regions of wild-type (wt) mice and were not observed in other regions (i.e. hippocampus, striatum and cortex) of the CNS in hSOD1[G93A] mice (Supplementary Fig. 1a shows a representative striatal region). NK cells isolated from the spleen and the spinal cord of wt and hSOD1[G93A] mice were analysed for the expression of activation markers and cytotoxic factors CD69, Granzyme b (GZMb) and perforin (PRF). The NK cells infiltrated in the CNS of hSOD1[G93A] mice showed higher frequency of GZMb and CD69, while no differences were observed in frequency and marker expression on NK cells isolated from the spleen of wt and

**Table 1 List of patients for NK cell staining in slices.**

| Patients | Clinical diagnosis | Gender | Site of onset | Age | Disease duration (months) | C9orf72 repeat expansion FUS and SOD1 mutation, TDP43 pathology |
|---|---|---|---|---|---|---|
| 1 | sALS | M | Spinal | 53 | 33 | C9orf72, TDP43 |
| 2 | sALS | M | Bulbar | 58 | 11 | TDP43 |
| 3 | sALS | M | Spinal | 66 | 22 | TDP43 |
| 4 | sALS | F | Spinal | 64 | 34 | C9orf72, TDP43 |
| 5 | sALS | M | Spinal | 53 | 33 | C9orf72, TDP43 |
| 6 | sALS | F | Bulbar | 74 | 31 | C9orf72, TDP43 |
| 7 | sALS | F | Bulbar | 66 | 35 | TDP43 |
| 8 | sALS | M | Spinal | 68 | 12 | TDP43 |
| 9 | sALS | M | Spinal | 69 | 24 | TDP43 |
| 10 | sALS | F | Bulbar | 65 | 24 | TDP43 |
| 11 | sALS | M | Bulbar | 40 | 35 | TDP43 |
| 12 | sALS | M | Spinal | 71 | 33 | TDP43 |
| 1 | Control | M | n/a | 54 | n/a | – |
| 2 | Control | M | n/a | 56 | n/a | – |
| 3 | Control | F | n/a | 66 | n/a | – |
| 4 | Control | F | n/a | 79 | n/a | – |
| 5 | Control | F | n/a | 35 | n/a | – |
| 6 | Control | M | n/a | 49 | n/a | – |
| 7 | Control | M | n/a | 75 | n/a | – |
| 8 | Control | M | n/a | 74 | n/a | – |

Site of onset (region in which first symptoms occurred): bulbar onset, spinal onset. Age: years. Disease duration: time from diagnosis until death in months. Fused in sarcoma/translocated in liposarcoma (FUS), chromosome 9 open reading frame 72 (C9orf72). The postmortem delay for ALS patients and controls was of 8 ± 2 h.
ALS Amyotrophic lateral sclerosis, sALS sporadic ALS, M male, F female, n/a not applicable.

**Table 2 List of patients for peripheral blood NK cell analyses.**

| Patients | Clinical diagnosis | Gender | Site of onset | Disease duration | Score |
|---|---|---|---|---|---|
| 1 | sALS | F | Spinal | 18 | C |
| 2 | sALS | M | Spinal | 9 | A |
| 3 | sALS | F | Spinal | 16 | A |
| 4 | sALS | M | Bulbar | 7 | C |
| 5 | sALS | F | Spinal | 98 | B |
| 6 | sALS | M | Bulbar | 8 | A |
| 7 | sALS | F | Spinal | 39 | C |
| 8 | sALS | M | Bulbar | 44 | C |
| 9 | sALS | M | Bulbar | 22 | C |
| 10 | sALS | F | Spinal | 24 | A |
| 11 | sALS | M | Spinal | 25 | B |
| 12 | sALS | M | Spinal | 50 | C |
| 14 | sALS | M | Bulbar | 16 | B |
| 15 | sALS | F | Bulbar | 12 | C |

Site of onset (region in which first symptoms occurred): bulbar onset, spinal onset.
ALS Amyotrophic lateral sclerosis, sALS sporadic ALS, M male, F female.
Disease duration: months from diagnosis to blood sample. Score: ALS patient deficit was scored according to the following scale: A—moderate; B—medium-severe; C—severe; D—complete.
The age of ALS patients was 58.37 ± 3.2 years. Healthy donors were recruited between 50.26 ± 10.2 years, of different gender ($n = 14$).

$hSOD1^{G93A}$ mice (Fig. 2b, c); PRF was expressed in all the NK cells. To better characterize the position of NK cells in the CNS, relative to vessels, we calculated the mean distance of $NKp46^+$ cells from $CD31^+$ (endothelial) cells, that was $152 \pm 29 \mu m$ (Fig. 2d), indicating that many NK cells observed were not in proximity of vessels (i.e. perivascular) but entered the brain parenchyma. Since the involvement of adaptive immune response in ALS is debated[20], we also investigated the infiltrating lymphocyte populations. Spinal cord and motor cortex of $hSOD1^{G93A}$ mice showed increased accumulation of $CD19^+$ B cells, $CD8^+$ and $CD4^+$ T cells, with a time dependent reduction of $CD8^+$ T cells at later stages (Supplementary Fig. 1b). To identify possible chemotactic signals involved in NK cell recruitment, we focused our attention on CCL2, a chemokine whose expression increases both in ALS patients and in $hSOD1^{G93A}$ mice[21] and plays a major role in NK cell recruitment[14]. At this aim, $hSOD1^{G93A}$ mice were treated (i.p.) with a CCL2 blocking antibody (Ab-CCL2) and data reported in Fig. 2e show that Ab-CCL2 treated mice had a significant reduction of NK cells in the CNS at early disease stages. In addition, Ab-CCL2 treatment decreased the number of $CD11b^+$ myeloid cells (motor cortex: Veh $1099 \pm 134$; Ab-CCL2 $341 \pm 120$; spinal cord: Veh $2169 \pm 271$; Ab-CCL2 $460 \pm 132$. $n = 2$–6 mice per condition), and the number of $CD3^+$ cells infiltrating the CNS in $hSOD1^{G93A}$ mice (motor cortex: Veh $53 \pm 26.3$; Ab-CCL2 $2.7 \pm 1.2$; spinal cord: Veh $22.3 \pm 7.8$; Ab-CCL2 $4 \pm 1.3$. $n = 3$–6 mice per condition). These low numbers hampered further subpopulation analysis. These results demonstrate that NK cells populating the CNS affected regions of $hSOD1^{G93A}$ mice are recruited to the brain by CCL2, directly or through other cell recruitment.

**NK cell depletion increases survival in mouse models of ALS $hSOD1^{G93A}$ and $TDP43^{A315T}$.** To assess the possible role of NK cells in disease onset and progression, we also used another animal model, the $TDP43^{A315T}$ mice, which express a mutated TAR-DNA binding protein, accumulate pathologic aggregates of ubiquitinated proteins in specific neurons, and activate local astrocytes and microglia, with the loss of both upper and lower motor neurons[22]. We treated 8-week-old $hSOD1^{G93A}$ and

$TDP43^{A315T}$ mice with an antibody against NK1.1, to deplete the NK cell population (see scheme in Fig. 2f). The efficacy of this treatment in depleting the NK cell ($CD3^+/NK1.1^-$) population was already shown[23], and preliminarily confirmed in these new experiments. $hSOD1^{G93A}$ and $TDP43^{A315T}$ mice depleted from NK cells showed a delay in onset of motor impairment and an increase in mean survival time in comparison with vehicle-treated mice (Fig. 2g), a delay in the onset of the paralysis (in $hSOD1^{G93A}$ mice, Fig. 2h), with no effect on disease progression. The timing of NK cell depletion is crucial, because when NK1.1 treatment of $hSOD1^{G93A}$ mice started at 13 weeks of age, no effects on survival time was observed ($hSOD1^{G93A}$ vehicle: $139.8 \pm 1.9$ days, $n = 5$; Ab-NK1.1: $136.8 \pm 1.6$ days, $n = 4$), in accordance with another study[24]. Behavioural tests to evaluate the changes in muscle strength and motor coordination show that NK cell-depleted mice preserved motor functions for a longer time, with no gender-related differences (Supplementary Fig. 2a, b). These results demonstrate the involvement of NK cells in the onset of motor signs and survival in two different mouse models of fALS, without effect on disease progression.

NK cell depletion did not alter the accumulation of $CD4^+$ and $CD8^+$ T cells either in motor cortex or spinal cord of $hSOD1^{G93A}$ mice (Supplementary Fig. 2c), so we can exclude that the observed effects are due to altered infiltration of these T cells.

**NK cells exert cytotoxic function against MNs in ALS.** To investigate the mechanisms of NK cell contribution to motor sign progression, we assayed their cytotoxic function in $hSOD1^{G93A}$ mice. We first analysed NKG2D ligands (*mult-1* and *rae-1*) and DNAM-1 (DNAX Accessory Molecule-1) ligands (*nectin-2* and *pvr*) expression in the spinal cord and motor cortex of $hSOD1^{G93A}$ mice: data in Fig. 3a, b show that $Smi32^+$ (a neuronal marker of neurofilament H) cells in the spinal cord co-stain with a murine ligand for NKG2D, Mult-1, only in $hSOD1^{G93A}$ and not in wt mice. In accordance, qRT-PCR analysis revealed that neurons ($NeuN^+$ cells), sorted from the spinal cord of $hSOD1^{G93A}$ mice, overexpressed *mult*-1 and *nectin*-2 but not *rae*-1 and *pvr*, in comparison with wt neurons (Fig. 3c). This selective overexpression may reflect the functional redundancy of NKG2D and DNAM-1 ligands[25]. Since NKG2D ligands are involved in triggering the cytotoxicity of NKG2D receptor-expressing NK cells, we performed a cytotoxic assay with primary neuronal cultures obtained from the lumbar region of the spinal cord of 13-week-old (early phase) $hSOD1^{G93A}$ and wt mice using, as effector cells, NK cells isolated from the spleen of wt mice (Fig. 3d). We first demonstrated that, also in primary cultures, $hSOD1^{G93A}$ $Smi32^+$ cells exhibited an increased expression of Mult-1 protein, which was absent in primary wt cultures (Supplementary Fig. 3a). Co-incubation of wt NK cells with primary spinal $hSOD1^{G93A}$ neurons resulted in an increased percentage of $CD107a^+$ (a lysosomal marker of active degranulation) NK cells, in comparison with wt neurons, across a variety of ratios (Fig. 3d and Supplementary Fig. 3b). Consistently, exposure to NK cells only affected the viability of $hSOD1^{G93A}$ neurons, not of wt neurons (Fig. 3d and Supplementary Fig. 3b).

To investigate whether the expression of mutated hSOD affected NK cell functionality, NK cells isolated from the spleen of $hSOD1^{G93A}$ mice were tested for cytotoxicity against neurons. The data shown in Supplementary Fig. 3c indicated that $hSOD1^{G93A}$ NK cells exhibited similar cytotoxic activity against wt or $hSOD1^{G93A}$ neurons, as shown for wt NK cells. To determine the involvement of NKG2D in the neurotoxic activity of NK cells, NK cell-neuron co-culture experiments were performed in the presence of anti-NKG2D blocking antibody; under these conditions the NK cell-mediated cytotoxic activity on

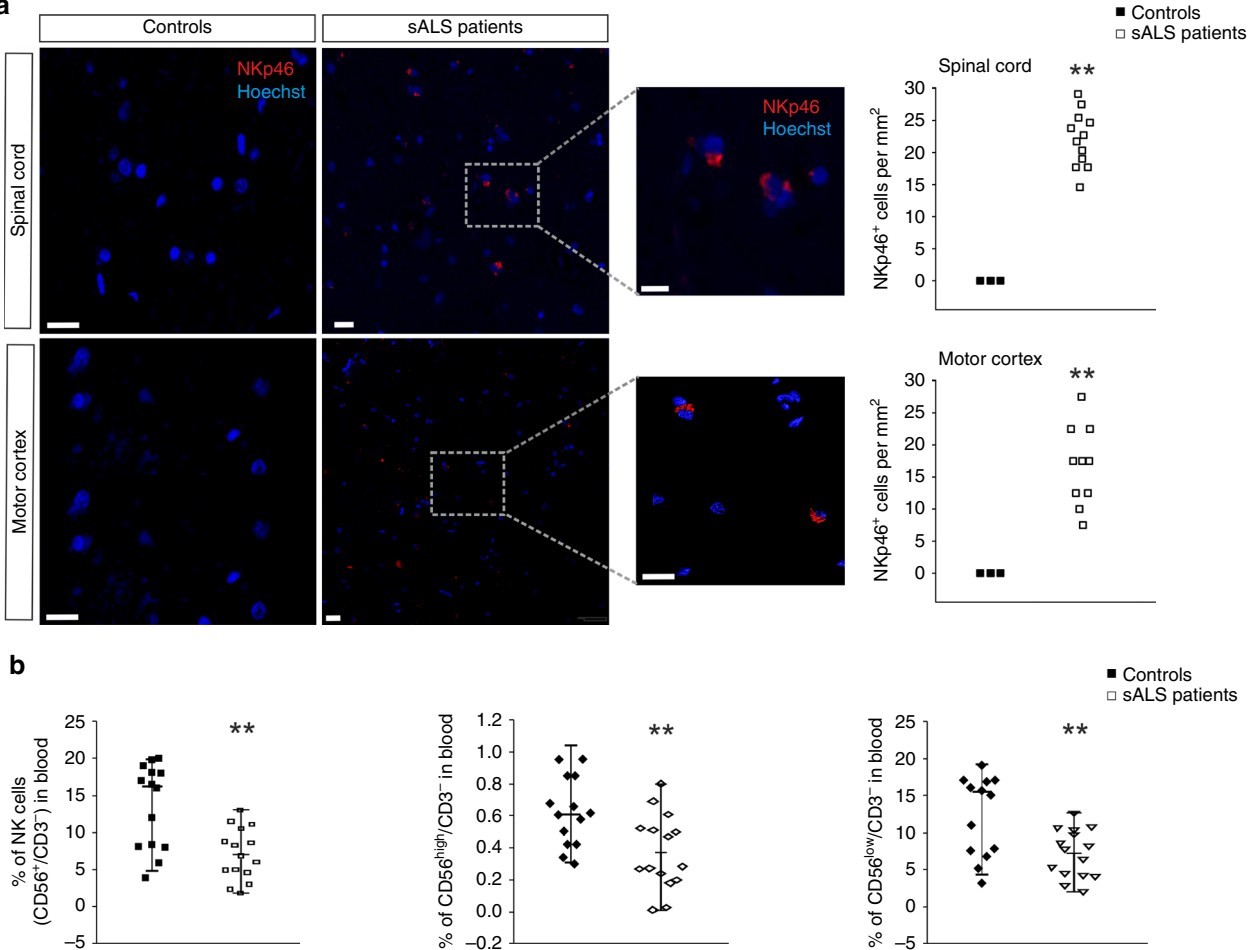

**Fig. 1 NK cells infiltrate the CNS affected regions in ALS patients. a** NKp46[+] cells (in red) in slices of the spinal cord and motor cortex obtained from sALS patients, not present in control patients. 3d reconstruction of one representative NK cell on the right. Scale bar: 20 μm. ($n = 8$ controls, 12 sALS patients **$P < 0.01$ power 0.999 two-tailed Student's $t$-test). Each dot represents the number of NKp46[+] cells per patient. Only 3 representative black squares are reported in the graph for Controls, in which were no detected NK cells. **b** Percentage of CD56[+/high/low]/CD3[−] cells obtained from peripheral blood of control or sALS patients ($n = 14$ controls, 15 sALS patients. **$P < 0.01$ power 0.860 two-tailed Student's $t$-test). Each dot represents the % of NK cells per patient. Error bars show mean ± SEM. Source data are provided as a Source Data file.

neurons was abolished (Fig. 3d). We then investigated the molecular mechanisms involved in the neurotoxic activity of NK cells, interfering with specific stages of lysosome exocytosis. At this aim we performed in vitro neurotoxicity assay in the presence of dynasore (dynamin inhibitor) or wiskostatin (N-WASP inhibitor). Both dynamin and N-WASP inhibition abolished NK cells-mediated hSOD1[G93A] neuron death (Fig. 3d). To evaluate the role of perforin in the lytic activity, NK cells isolated from *prf1* ko mice were incubated, in a cytotoxicity assay, with wt or hSOD1[G93A] neurons. Results shown in Fig. 3d demonstrated that the absence of perforin abolished NK cells-mediated hSOD1[G93A] neuron death.

To further investigate in vivo the possible cytotoxic activity of NK cells against MNs, we verified the presence of potential physical contacts between NK cells and MNs in the spinal cord of hSOD1[G93A] mice, as shown in Fig. 3e. We then analysed the viability of MNs in the ventral horns of the lumbar spinal cord of hSOD1[G93A] mice and observed a reduction of their number in comparison with wt mice, already at 13 weeks (Fig. 3f). This effect was attenuated in NK cell-depleted hSOD1[G93A] mice, where a higher number of MNs (Smi32[+] and ChAT[+] cells) was observed,

in comparison with vehicle-treated hSOD1[G93A] mice (Fig. 3f and Supplementary Fig. 3d). At later stages (16 weeks), the spinal cord of NK cell-depleted hSOD1[G93A] mice showed the same number of MNs, as compared with vehicle-treated hSOD1[G93A] mice, indicating that the effect of NK cell depletion has a limited time window of efficacy (Supplementary Fig. 3e). Similar results were observed in male hSOD1[G93A] mice soon after disease onset (Supplementary Fig. 3f). The role of NKG2D in NK cell-mediated MN death was further demonstrated knocking down NKG2D, treating mice with dendrimers-loaded with NKG2D-specific siRNA. The efficacy of siRNA silencing was verified in vitro on human and murine NK cells: Supplementary Fig. 4a shows siRNA uptake by NK cells after 48 h and the effects on *klrk1* gene and NKG2D protein expression is shown in Supplementary Fig. 4b, c. Functionally, dendrimer-siRNA NKG2D-transfected NK cells have a reduced cytotoxic activity against the murine tumour cells GL261 (Supplementary Fig. 4d). In vivo, in hSOD1[G93A] mice, siRNA treatment reduced the circulating NKG2D[+] NK cells by 26% ± 5.5%, and silenced NKG2D expression (MFI) by 29% ± 5.5% in cells that remained NKG2D-positive (Supplementary Fig. 4e). Motor neuron count, in the spinal cord of siRNA-loaded

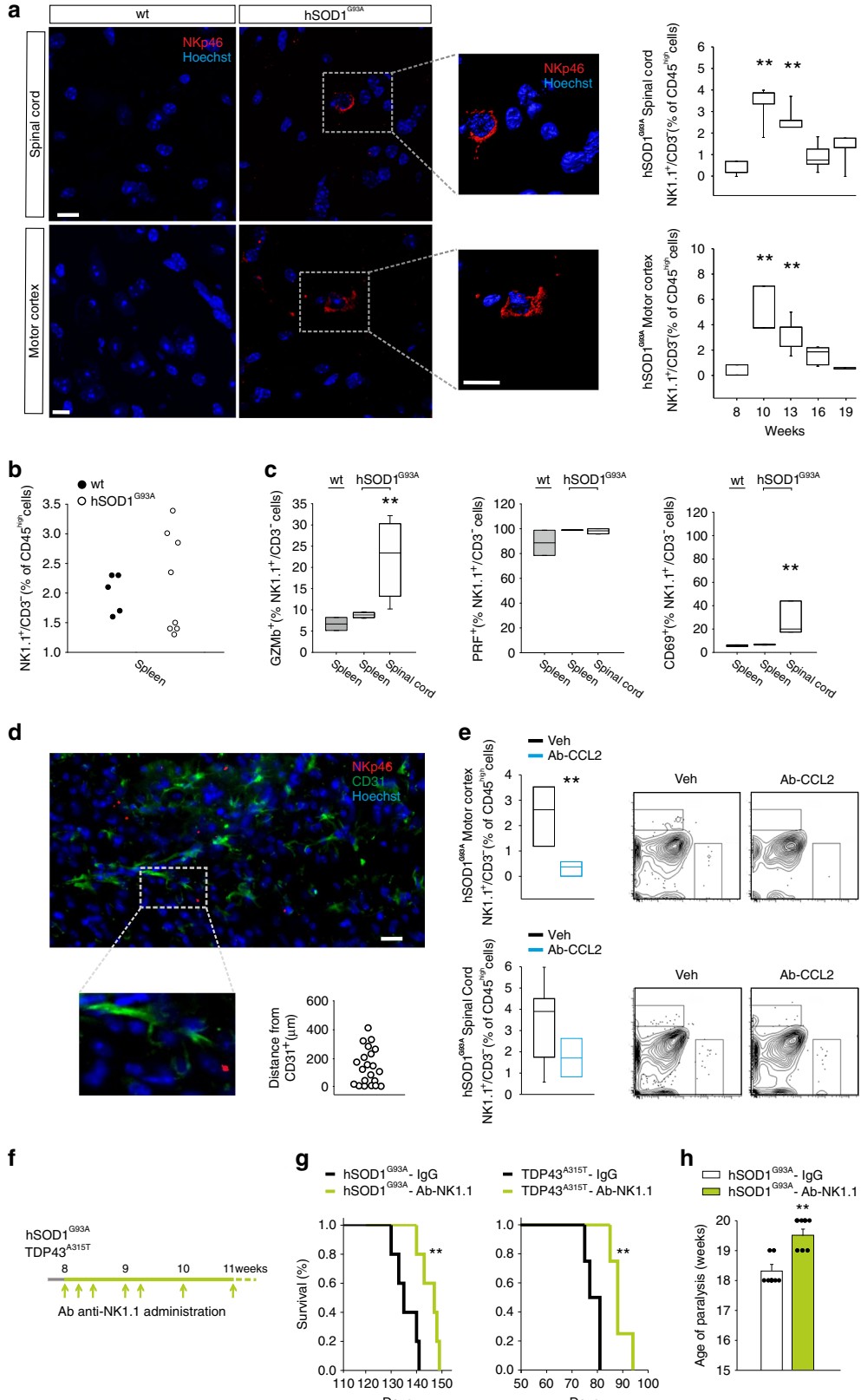

dendrimer-treated mice, revealed increased numbers in comparison with empty or scrambled siRNA-dendrimer-treated mice (Fig. 3f).

Taken together, these data demonstrate the cytotoxic activity of NK cells against motor neurons expressing NKG2D ligands in the hSOD1[G93A] mouse model.

**NK cell depletion induces a protective microglia phenotype.** We previously demonstrated that NK cells modulate microglia phenotype in the context of glioma[16]. To investigate the possible cross talk between NK cells and microglia in hSOD1[G93A] mice, we analysed the microglia phenotype upon NK cell depletion. To this end, we analysed Iba[+] (a microglia/macrophage marker[26])

**Fig. 2 NK cells infiltrate the CNS and affect survival time in ALS models. a** Left: NKp46$^+$ cells (in red) in cerebral slices of the lumbar spinal cord and motor cortex of hSOD1$^{G93A}$ mice (13 weeks), not present in wt mice. Scale bar: 10 μm. 3d reconstruction of one representative NK cell on the right. Right: time course analysis of NK1.1$^+$/CD3$^-$ cells in the lumbar spinal cord and the cerebral motor cortex of hSOD1$^{G93A}$ mice (mice at 8 and 19 weeks, $n = 4$; mice at 10, 13 and 16 weeks, $n = 6$. **$P < 0.01$, power 0.999, one-way ANOVA $vs$ 8 weeks). **b** Frequency of NK1.1$^+$/CD3$^-$ cells in the spleen of wt and hSOD1$^{G93A}$ mice (wt, $n = 5$; hSOD1$^{G93A}$ $n = 8$). **c** Expression of GZMb, PRF and CD69 in the spleen and spinal cord of wt and hSOD1$^{G93A}$ mice (wt, $n = 3$; hSOD1$^{G93A}$, $n = 5$, **$P < 0.01$, power 0.841, one-way ANOVA). **d** Distances of NKp46$^+$ cells from vessel endothelial cells (CD31$^+$ cells) in the spinal cord of hSOD1$^{G93A}$ (13 weeks) mice ($n = 4$ mice/21 cells). Scale bar: 20 μm. Representative immunofluorescence is shown. **e** NK1.1$^+$/CD3$^-$ cells frequency in the motor cortex (top) and in the spinal cord (bottom) of hSOD1$^{G93A}$ mice treated with Ab-CCL2 ($n = 6$, *$P < 0.05$ power 0.916 two-tailed Student's $t$-test). Representative density plots are shown on the right. **f** Scheme of Ab-NK1.1 administration. **g** Kaplan–Meier curve of hSOD1$^{G93A}$ and TDP43$^{A315T}$ mice treated with vehicle or Ab-NK1.1 (hSOD1$^{G93A}$, vehicle: 136.8 ± 1.7 days; Ab-NK1.1: 147.2 ± 2.4 days; $n = 5$. TDP43$^{A315T}$, vehicle: 78.5 ± 1.5 days; Ab-NK1.1: 88.7 ± 1.9 days, $n = 4$. Data are the mean ± s.e.m. **$P < 0.01$, two-sided log-rank test). **h** Age of paralysis, calculated as the absence of hindlimb movement, upon Ab-NK1.1 treatment in hSOD1$^{G93A}$ mice; vehicle: 18.3 ± 0.2 weeks; Ab-NK1.1: 19.5 ± 0.2 weeks ($n = 7$, data are the mean ± s.e.m. **$P < 0.01$ power 0.921 two-tailed Student's $t$-test. For boxplots (**a, c, e**), the center line, boxes and whiskers represent the median, inner quartiles, and rest of the data distribution, respectively.

cells in spinal cord tissue by two-photon microscopy, measuring cellular branching and the total area covered by the single cells. Our data show that hSOD1$^{G93A}$ microglia, in NK cell-depleted mice, have reduced soma area, increased length of branches and, in general, cover a wider parenchymal region, compared to hSOD1$^{G93A}$ microglia from vehicle-treated mice (Fig. 4a). Moreover, NK cell-depleted hSOD1$^{G93A}$ mice had less Iba1$^+$ cells, in the ventral horns of the spinal cord (Fig. 4b). To further study NK cell-microglial communication, we isolated microglial cells from the spinal cord of wt and hSOD1$^{G93A}$ mice, treated with Ab-NK1.1 or vehicle at 13 or 16 weeks and analysed the expression of a number of genes that characterize microglia phenotype in ALS[22]. Microglia from hSOD1$^{G93A}$ mice preferentially expressed genes involved in inflammatory microenvironment, including *il-6, il-1β, tnf-α* and *nos2* (Fig. 4c) and ROS production (*nox2, p47phox*) (Fig. 4d). In NK cell-depleted hSOD1$^{G93A}$ mice, the expression of these genes was significantly reduced (Fig. 4c, d), with the simultaneous increase of expression of the anti-inflammatory markers *chil3, arg-1*, and *tgfβ*, of the ROS scavenger *msod1* (Fig. 4e, f) and the modulation of other genes associated with a homeostatic neuroprotective microglial phenotype (*p2yr12, trem2, kcnn4, bdnf, il-15*) (Fig. 4g). We did not observe gender differences in these experiments (Supplementary Fig. 5a) and no alteration in *hsod1* expression was observed (hSOD1$^{G93A}$ vehicle: 1 ± 0.34; Ab-NK1.1: 0.87 ± 0.09; $n = 5$). At late-stage disease, microglia isolated from NK cell-depleted hSOD1$^{G93A}$ mice showed only minor differences in gene expression as compared with untreated hSOD1$^{G93A}$ mice (*il-6, arg1* and *bdnf* Supplementary Fig. 5b).

**IFNγ mediates NK cell activities in ALS**. Investigating the potential mediators of NK cell effects in ALS, we focused our attention on IFNγ produced by NK cells. We compared the expression level of IFNγ in the spinal cord of hSOD1$^{G93A}$ and wt mice, and detected higher levels of IFNγ in hSOD1$^{G93A}$, in comparison with wt mice (Fig. 5a); NK cell depletion significantly reduced the cytokine levels (Fig. 5a). In accordance, in hSOD1$^{G93A}$ mice, the frequency of NK cells infiltrating in the spinal cord and expressing IFNγ was 14.25% ± 2.68% (Fig. 5b) higher than what reported in brain tumour[16]. To investigate the possible relevance of IFNγ produced by NK cells, we treated mice with the IFNγ-blocking antibody XMG1.2 (see scheme in Fig. 5c). The efficacy of XMG1.2 treatment was previously shown in mouse brain[16], and was verified in the spinal cord by MHCII expression analysis in microglia and antibody staining (Supplementary Fig. 6a, b).

We observed that IFNγ immunodepletion increased the mean survival time of hSOD1$^{G93A}$ mice (Fig. 5d) and delayed paralysis

onset (Fig. 5e) without affecting disease progression. Mice were longitudinally evaluated for muscle strength, locomotor activity and motor coordination: IFNγ-depleted mice showed better score in the hindlimb extension reflex, in the inverted grid and in the Rotarod tests (Supplementary Fig. 6c). Blocking IFNγ had consequences similar to NK cell depletion on microglial phenotype, modulating inflammatory gene expression (Fig. 5f,g).

Since IFNγ stimulates the neuronal release of CCL2[27] which in our mouse model contributes to NK cell accumulation in the CNS (shown in Fig. 2e), we investigated the neuronal expression of *ccl2* in NeuN$^+$ cells isolated from the spinal cord of hSOD1$^{G93A}$ mice, and observed an increased expression in comparison with wt (Fig. 5h). The expression of *ccl2* in NeuN$^+$ cells was abolished in neurons isolated from the spinal cord of NK cell- and IFNγ-depleted hSOD1$^{G93A}$ mice (Fig. 5h). At difference, microglial *ccl2* expression increased in NK cell-depleted hSOD1$^{G93A}$ mice compared to vehicle hSOD1$^{G93A}$ and wt mice (Fig. 5i).

**NK cells and IFNγ affect Treg number in hSOD1$^{G93A}$ mice**. IFNγ has a crucial role in driving regulatory T cell (Tregs) differentiation and proliferation[28] and Treg become reduced and dysfunctional in ALS patients[10]. We performed an immunofluorescence analysis of FOXP3$^+$ cells (Treg cells) in the spinal cord of NK cells-depleted hSOD1$^{G93A}$ mice and observed an increased number of FOXP3$^+$ cells in comparison with control mice (Fig. 6a). This was confirmed by mRNA analysis, where *foxp3* expression, in the spinal cord of hSOD1$^{G93A}$ mice, was significantly increased upon NK cell depletion (Fig. 6b). In line with these results, blocking IFNγ also increased *foxp3* expression demonstrating the role of this cytokine in suppressing Treg cells in hSOD1$^{G93A}$ mice (Fig. 6b).

**NK cell-motor neuron interactions in ALS patients**. To investigate the possible interaction among MNs and NK cells also in humans, we determined the expression of NKG2D ligands in the spinal cord and the cerebral motor cortex MNs of sALS patients and found immunoreactivity for UL16 binding protein 3 (ULBP3), which colocalized with Smi32$^+$ cells and was completely absent in controls (Fig. 7a, b). Furthermore, Fig. 7c shows the occurrence of physical contacts between NK cells and MNs in the spinal cord of ALS patients. In these same tissues, the mean distance of NKp46$^+$ cells from vessels (CD31$^+$ cells) is 188 ± 43 μm, Fig. 7d), indicating cells in brain parenchyma. Similarly to what observed in hSOD1$^{G93A}$ mice, higher IFNγ levels were detected in the spinal cord of sALS patients in comparison with controls (Fig. 7e), and more IFNγ$^+$ NK cells were present in the peripheral blood (Fig. 7f). On the other hand, no differences were detected in the cytotoxic activity of NK cells isolated from the peripheral

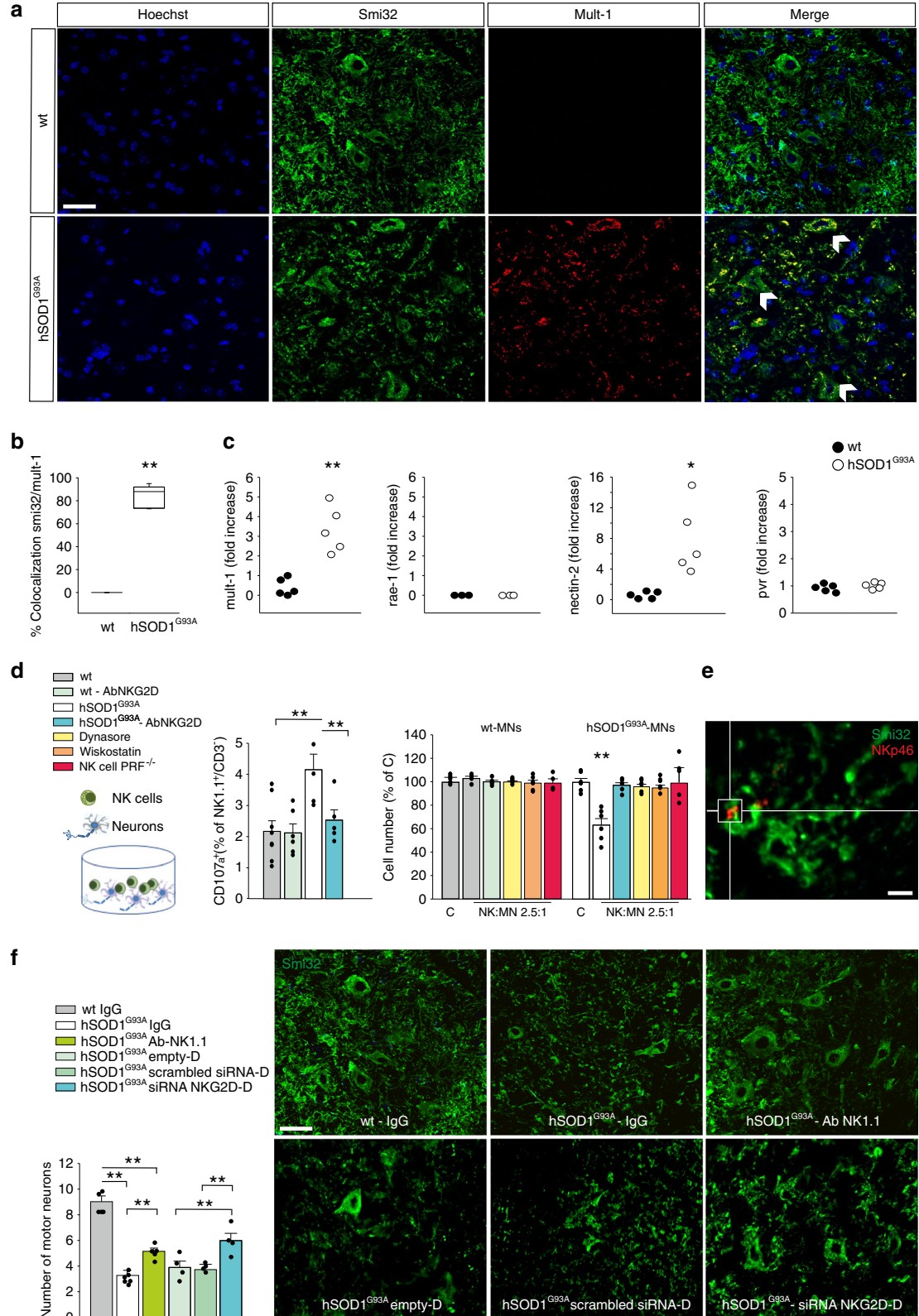

blood of sALS patients and controls, seen as CD107a expression upon interaction with NK-susceptible tumor (K562) target cells (Fig. 7g), and also the expression of GZMb and PRF did not differ (Fig. 7h, i).

## Discussion

We identified a new potential mechanism to explain the non-cell-autonomous motor neuron degeneration which occurs in ALS. We demonstrated that, in sALS patients and in the mouse

**Fig. 3 NK cell–MN communication in ALS. a** Expression of Mult-1 in the spinal cord of wt and hSOD1[G93A] mice (13 weeks) (n = 5). Scale bar: 20 μm. **b** Colocalization of Smi32 and mult-1 covered area in spinal cord of wt and hSOD1[G93A] mice (n = 5 data are the mean ± s.e.m. **P < 0.01 power 1 two-tailed Student's t-test). For boxplots, the center line, boxes and whiskers represent the median, inner quartiles, and rest of the data distribution, respectively. **c** RT-PCR of mult-1, rae-1, nectin-2 and pvr genes in NeuN[+] cells sorted from lumbar spinal cord of wt and hSOD1[G93A] mice (13 weeks; n = 5, *P < 0.05, **P < 0.01 power >0.85 vs wt mice, one-way ANOVA). **d** NK cells, isolated from the spleen of wt mice, were incubated with primary cultures of neurons obtained from the spinal cord of wt or hSOD1[G93A] (13 weeks) mice (left), in the presence of a specific Ab against NKG2D, of dynasore, wiskostatin or vehicle; degranulation of NK cells (having subtracted basal degranulation) in a 2.5:1 ratio with neurons was assessed by FACS analysis of CD107a[+] cells (middle panel); cell viability in neuron cultures is shown in the right panel (wt, n = 7. hSOD1[G93A], vehicle n = 6, treated with AbNKG2D, dynasore and wiskostatin n = 4, PRF[−/−] n = 4, **P < 0.01 power 0.999 one-way ANOVA). Error bars show mean ± SEM. **e** Representative image of NK cell—MN contacts in the spinal cord of hSOD1[G93A] mice (original magnification, ×600). Scale bar: 20 μm. (hSOD1[G93A] mice, n = 6) **f** Quantification of MNs (counted as Smi32[+] cells in the ventral horns of the spinal cord) in wt (n = 5) and hSOD1[G93A] mice treated with vehicle (n = 6) or Ab-NK1.1 (n = 5), and dendrimer or dendrimer-loaded with scrambled siRNA or siRNA-NKG2D[41] (n = 4). Data are shown as mean ± s.e.m. **P < 0.01 power >0.9 one-way ANOVA). Representative immunofluorescence images are shown on the right. Scale bar: 20 μm. Source data are provided as a Source Data file.

model hSOD1[G93A], NK cells infiltrate the CNS affected regions. We observed that NeuN[+] cells from the spinal cord of hSOD1[G93A] mice expressed increased levels of the chemokine CCL2, and that CCL2 neutralization reduced CNS infiltration by NK cells. In addition, NK cell depletion increased survival in hSOD1[G93A] and TDP43[A315T] mice and modulated the activation of microglia and the infiltration of Treg cells in hSOD1[G93A] mice. Finally, we reported that both motor neurons of ALS patients and those of the hSOD1[G93A] mice express higher level of NK2GD ligands, important for NK cell cytotoxic activity. These observations support the knowledge of ALS as a non-cell-autonomous disease, where motor neuron degeneration results from communication defects with immune and glial cells.

Previous reports have described that NK cells are present in the spinal cord of end-stage mSOD1 mice[29], and increased NK cell number is reported in the blood of ALS patients[10]. At difference, we considered the relative amount of NK cells with respect to other populations of PBMC and demonstrated that the frequency of NK cells in peripheral blood of ALS patients is reduced. Even if a direct relation among NK cells number and their frequency is not obvious, further experiments should be performed to establish possible correlations with disease stage at the moment of blood collection, also considering ongoing pharmacological therapies. However, the function of NK cells in disease progression had never been addressed. We demonstrated that NK cell recruitment in the CNS of hSOD1[G93A] mice is CCL2-dependent; however, we cannot discriminate among a direct effect of CCL2 on NK cells and an indirect effect, coordinating the recruitment of other cells[30] (such as monocytes) and the production of additional chemoattractant (such as CX₃CL1)[12,13]. The neuronal production of CCL2 has been associated with neuronal injury in different models of trauma, pain or viral infection[31–33], and CCL2 level increases in the spinal cord tissue of SOD1[G93A] mice[29,30]. These data identify CCL2 as an early damage signal of neural tissue.

Motor neuron degeneration in ALS has been correlated with the presence of an uncontrolled inflammatory environment[34,35] or with an inappropriate microglia activity, which impairs axonal regeneration[36,37]. Our results point toward the importance of re-educating microglia toward a homeostatic phenotype to reduce motor neuron loss and identify NK cells as possible therapeutic targets. While Treg and microglia modulation correlates with disease progression in mutant SOD1 mice and ALS patients[3,21,30,38], we show that NK cell-mediated modulation of these cell types only affects survival and onset time, possibly due to specific or partial alteration of microglia or Treg phenotype in the different experimental settings.

A new paradigm that emerged from our data is that motor neuron degeneration results from direct cell-to-cell contacts with NK cells, with mechanisms involving granule exocytosis and perforin expression. A recent report on a model of peripheral nerve injury, describes that the NKG2D ligand Rae1 is upregulated in the lumbar DRG of adult mice upon cutting the corresponding spinal nerve, and that neurite fragmentation is boosted by IL-2 stimulated NK cells[17]. Our data that both in CNS tissue of ALS patient and in hSOD1[G93A] mice, motor neurons express different NKG2D ligands, identify NKG2D on NK cells as possible molecular targets to reduce motor neuron loss. Furthermore, the observation that hSOD1[G93A] NK cells do not exert cytotoxic activity toward wt motor neurons demonstrate that the kill-me signal comes from mutant motor neurons.

Recent data identify Treg cells as suppressor of microglia toxicity in mSOD[39,38] mice; and ALS patient infusion with expanded autologous Treg slows disease progression[18]. In addition to a direct neurotoxic activity of NK cells, we demonstrate an IFNγ dependent recruitment of Foxp3[+] Treg cells in the spinal cord of hSOD1[G93A] mice, which could explain the positive effects of NK cell depletion on mice survival[29] and microglia phenotype[9]. The mechanisms of Treg and NK cells cross talk in hSOD1[G93A] mice are not known, but could be mediated by IFNγ[38,39], since IFNγ blockade also increased foxp3 expression in the spinal cord, or by TGFβ[40]. In contrast, NK cell depletion did not alter CD4[+] or CD8[+] T cell infiltrate, although we cannot exclude a possible cytotoxic role of CD8[+] T cells in ALS[41,42], also as source of IFN-γ.

Altogether, our data support the critical role of innate immunity in ALS-associated neurodegeneration and demonstrate the neurotoxic properties of NK cells in the hSOD1[G93A] mice model. Targeting NK cells could revert the vicious cycle among NK cells (producing IFNγ) and neurons (producing CCL2), re-boosting the homeostatic functions of microglia and sustaining Treg cell recruitment in the spinal cord (see graphic summary in Fig. 8). Together with the data obtained in ALS patients, our results point to innate immunity as a key co-factor in this neurodegenerative disease. It remains to be investigated whether a direct neurotoxic role of NK cells could be relevant to other neurodegenerative diseases.

## Methods

**Materials**. All cell culture media, foetal bovine serum (FBS), goat serum, penicillin G, streptomycin, Na pyruvate, glutamine, the Thermo Script RT-PCR System, secondary Abs, and Hoechst (catalogue #33342, RRID:AB_10626776) were from GIBCO Invitrogen (Carlsbad, CA, USA). Glucose, Percoll, Papain (#P3125), phosphate buffered-saline (PBS) tablet (#P4417), Bovine Serum Albumine (BSA) and deoxyribonuclease I, were from Sigma-Aldrich (Milan, Italy). Chloral hydrate (#334085) was from Carlo Erba (Milan, Italy). NKp46 (M20) (#sc-18161, RRID: AB_2149152) antibody (Ab) was from Santa Cruz biotechnology (Santa Cruz, CA). ChAT (#AB144P) Ab was from Merck Millipore (Milan, Italy). BDNF was from Immunological Sciences (Rome, Italy). IFNγ ELISA kit (KAC1231), and Mult-1 Ab (Cat# 12-5863-81, RRID:AB 1210756) were from Life technologies Invitrogen (Carlsbad, CA, USA). Microbeads CD11b[+] were from Miltenyi Biotec (Bologna,

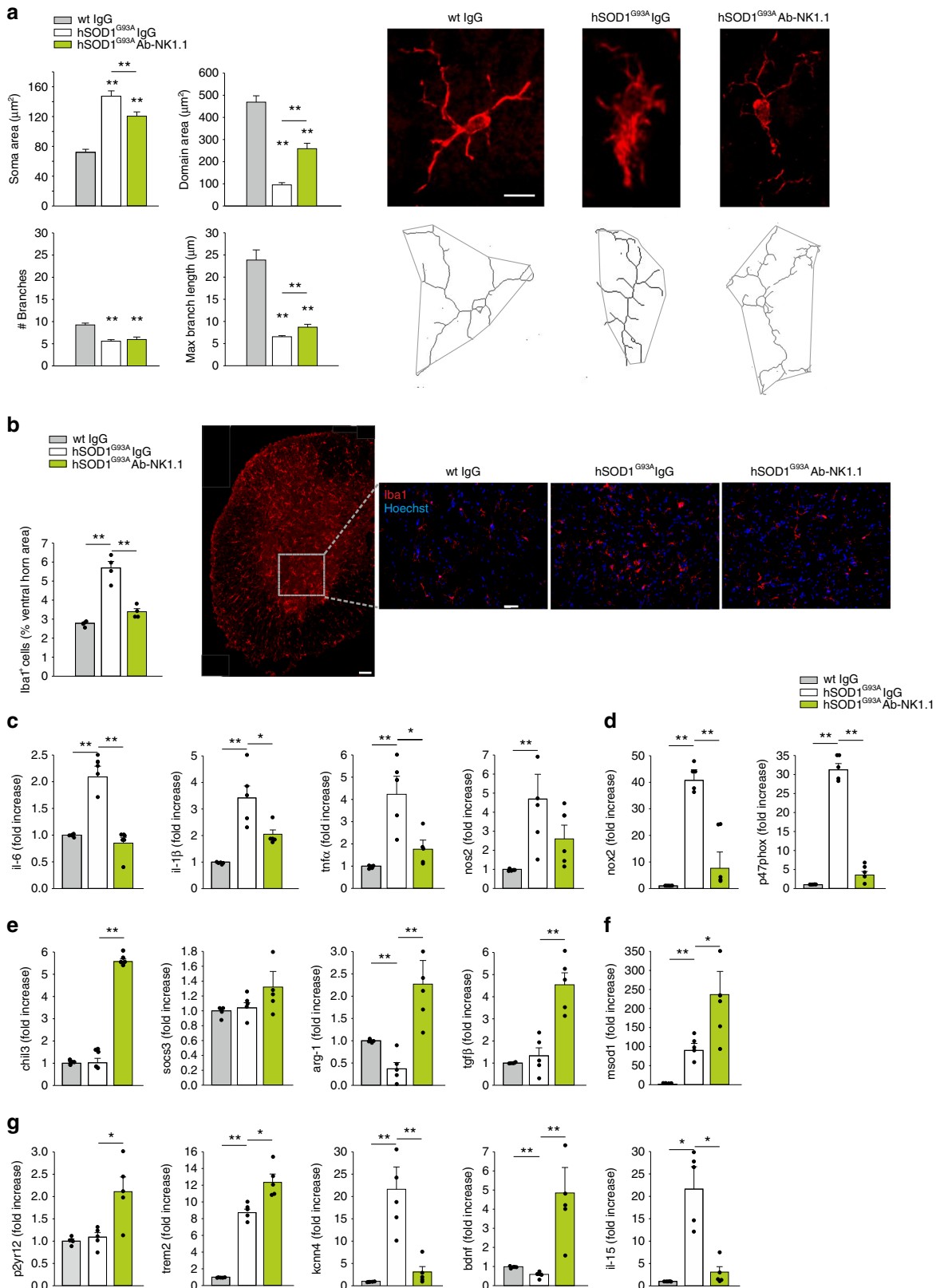

Italy). RNeasy Mini Kit was from Qiagen (Hilden, Germany). CD45, CD69, CD3, CD4, CD8, CD107a, CD133, NK1.1 Abs, and IL-15 were from eBioscience Inc., (San Diego, CA). Rabbit anti-Iba1 from Wako (Cat# 019-19741, RRID:AB_839504) (VA, USA). Rat anti-IFNγ monoclonal antibody (clone: XMG1.2) (Cat# BE0055, RRID:AB 1107694), anti-CCL2 (clone: 2H5) (Cat# BE0185, RRID:AB_10950302), and anti-NK1.1 (Cat# BE0036, RRID:AB_1107737) were from Bioxcell (West Lebanon, USA). ULBP-3 Ab (Cat# MAB1517, RRID:AB_2272829) was from R&D

systems (Minneapolis, MN, CANADA). Foxp3 Ab (Clone: FJK-16s) (Cat# 53-5773-82, RRID:AB_763537) was from Thermo Fisher (San Diego, CA). CD31 Ab (#3568 S, RRID:AB_10694616) was from Cell Signaling (WZ Leiden, The Netherlands). SMI-32 Ab was from BioLegend (San Diego, CA). siRNA-NKG2D ALEXA 488 (COD A10433) and scrambled siRNA were from Life technologies Invitrogen (Carlsbad, CA, USA). Dendrimers were synthesized for siRNA delivery as in[43].

**Fig. 4 NK cells modulate microglia phenotype in hSOD1$^{G93A}$ mice. a** Quantification of soma and scanning domain area, branch numbers and length of Iba1$^+$ cells in slices obtained from the spinal cord of wt and hSOD1$^{G93A}$ mice (13 weeks) treated with vehicle or Ab-NK1.1. Right: representative images (original magnification, ×600; Scale bar: 10 μm) of the maximal intensity projections of confocal z-stack images of Iba1$^+$ cells in the different conditions, converted to binary images and then skeletonized. ($n = 3$ mice per treatment, 30 cells per mice in at least six slices. **$P < 0.01$ power 0.921 one-way ANOVA). Error bars show mean ± SEM. **b** Mean (±s.e.m.) area of Iba1$^+$ cells (expressed as % of spinal cord area) in wt or hSOD1$^{G93A}$ mice (13 weeks) treated with vehicle or Ab-NK1.1 ($n = 4$, **$P < 0.01$ power 1 one-way ANOVA). Right: representative immunofluorescence image of Iba1$^+$ cells in spinal cord sections. Scale bar: 50 μm. **c–g** RT-PCR of il-6, il-1β, tnfα, nos2, nox2, p47phox, chil3, socs3, arg-1, tgfb, msod1, p2yr12, trem2, kcnn4, bdnf, il-15 gene expression in microglial cells isolated from the lumbar spinal cord of wt or hSOD1$^{G93A}$ mice (13 weeks) treated with vehicle or Ab-NK1.1 ($n = 5$ mice. Data are the mean ± s.e.m. *$P < 0.05$ **$P < 0.01$ power >0.85 one-way ANOVA). Source data are provided as a Source Data file.

**Animals**. Experiments described in the present work were approved by the Italian Ministry of Health (authorization n. 78/2017-PR) in accordance with the guidelines on the ethical use of animals from the European Community Council Directive of September 22, 2010 (2010/63/EU), and from the Italian D.Leg 26/2014. All possible efforts were made to minimize animal suffering, and to reduce the number of animals used per condition by calculating the necessary sample size before performing the experiments. Male and female hSOD1$^{G93A}$ [B6.Cg-Tg(SOD1-G93A) 1Gur/J line, the transgene copy number is typically 17-23] (RRID: IMSR_JAX:004435), TDP43$^{A315T}$ [B6.Cg-Tg(Prnp-TARDBP*A315T)95Balo/J] (Cat# JAX:010700, RRID:IMSR_JAX:010700), and C57BL/6-Prf1 < tm1Sdz > /J (Cat# JAX:002407) transgenic mice were obtained from Jackson Laboratory (Bar Harbor, ME, USA) and from Charles River (Calco, Italy). hSOD1$^{G93A}$ were also maintained as hemizygotes by breeding transgenic males with wild-type C57BL/6 J females from Charles River Laboratories, both maintained on C57BL/6 J genetic background. Age-matched non-transgenic C57BL/6 J mice were always used as controls. Transgenic mice were identified by PCR on DNA obtained from tail biopsies. Briefly, tail tips were digested (overnight, 58 °C) in a buffer containing 100 mM Tris–HCl pH 8, 0.1% SDS 20, 5 mM EDTA pH 8, 200 mM NaCl and 20 mg/ml proteinase K (Ambion-Thermo Fisher, Germany, #2548) and the genomic DNA was amplified with SsoFast Eva Green Supermix (Bio-Rad, California, #172-5201) using the following primers: SOD1 forward 5′-CATCAGCCC TAATCCATCTGA-3′; SOD1 reverse 5′- CGCGACTAACAATCAAAGTGA-3′. Mice were housed in standard breeding cages at a constant temperature (22 ± 1 °C) and relative humidity (50%), with a 12:12 h light:dark cycle (light on 07.00–19.00 h). Food and water were available ad libitum. Microbiological analyses were routinely (each 3–4 months) performed and defined endemic Norovirus and Helicobacter in our conventional animal facility. Mice were evaluated for motor deficits with a behavioural score system: 0 = Full extension of hind legs away from the lateral midline when the mouse is suspended by tail; the mouse must hold this position for 2 s, and is suspended 2–3 times; 1 = Collapse or partial collapse of leg extension towards lateral midline (weakness) or trembling of hind legs during tail suspension; 2 = Curling of the toes and dragging of at least one limb during walking; 3 = Rigid paralysis or minimal joint movement; foot not used for forward motion; 4 = Mouse cannot stand up in 20 s from either side, euthanasia.

**Patients enrolment**. We studied 15 patients with sporadic (mean age 61.3 ± 5.2 years), probable (clinically or laboratory supported) or definite ALS[44], recruited from the Rare Neuromuscular Diseases Centre of Umberto I Hospital in Rome (ref 3314/25.09.14, protocol n. 1186/14). The study was conducted according to the declaration of Helsinki and was approved by the institutional ethics committee. Informed consent was obtained from all the subjects. Patients were evaluated using the ALS Functional Rating Scale-Revised (ALSFRS-R), and the Medical Research Council (MRC) score. All patients took Riluzole 50 mg BID. Healthy Donors stratified for age and sex were also enrolled as control. Table 2 resume the characteristics of ALS patients and healthy donors enrolled in the study.

**Human brain and spinal cord tissues**. Postmortem material was obtained at autopsy from 12 ALS patients at the department of (Neuro)Pathology of the Amsterdam UMC, Academic Medical Center, (University of Amsterdam, the Netherlands). All patients fulfilled the diagnostic criteria for ALS (El Escorial criteria[45]) as reviewed independently by two neuropathologists. All patients with ALS died from respiratory failure. Control spinal cord tissue was obtained from eight patients who had died from a non-neurological disease. Both ALS and control patients included in the study displayed no signs of infection before death. Informed consent was obtained for the use of brain tissue and for access to medical records for research purposes and approval was obtained from the relevant local ethical committees for medical research. All autopsies were performed within 10 hours after death. ALS sporadic patients did not showed mutation for SOD or FUS proteins. Four patients were positive for the C9ORF72 hexanucleotide repeat expansion.

Table 1 resume the characteristics of ALS patients and control enrolled in the study.

Tissue preparation: paraffin-embedded tissue was sectioned at 6 μm and mounted on pre-coated glass slides (StarFrost, Waldemar Knittel Glasbearbeitungs GmbH, Braunschweig, Germany). Representative sections of all specimens were processed for haematoxylin and eosin, Klüver-Barrera and Nissl stains.

**Primary spinal cord neurons culture from adult mice**. Neurons cultures were obtained from spinal cord 13w C57BL6/J or hSOD$^{G93A}$ mice. Mice were euthanized by cervical dislocation and the spinal cords were removed; tissues were digested with 30U/ml papain at 37 °C. After 20 min the reaction was stopped by adding papain inhibitor and pipetting 10–15 times with a glass Pasteur pipette, until no more cell aggregates were visible. Cells were filtered with 100μm cellstrainer and centrifuged at $300 \times g$ for 8 min. The pellet was resuspended in complete Neurobasal medium (Gibco, part of Thermo Fisher, Germany #10888-022) (2 mM glutamine, 1% B27, 100 U/ml penicillin and 0.1 mg/mL streptomycin, BDNF 10 ng/ml) and plated ($2 \times 10^5$ cells/well) onto poly-L-lysine coated glass cover slips. After 14 days in culture, purity of neuronal cells (analysed as Smi32$^+$ cells) ranges between 50 and 60%.

**Immunofluorescence and FACS analysis**. Immune cells from wt or hSOD$^{G93A}$ mice were enriched by centrifugation on percoll 40%, washed in PBS and immunostained with fluorochrome-conjugated anti-CD3, anti-NK1.1, anti-CD19 and anti-CD45.2 to identify T cells (CD3$^+$NK1.1$^-$) NK cells (NK1.1$^+$CD3$^-$) and B cells (CD19$^+$) and analyzed by flow cytometry using a FACSCanto II (BD Biosciences). Data were elaborated using FlowJo Version 9.3.2 software (TreeStar).

**NK cells purification from blood patients**. Peripheral blood mononuclear cells (PBMCs), freshly isolated by lymphoprep (Nycomed AS, Oslo, Norway) were frozen and stored at −80 °C for up to 2 months. The day before the experiment, cells were thawed and kept in culture O/N for recovering. Then, death cells were removed, and cells were stained with NK cell subset specific antibodies or were used for functional assays. To determine NK cell degranulation potential, PBMCs were co-cultured with K562 cells at 1:1 effector/target ratio for 3 h in complete RPMI 1640 medium (Hepes 10 mM + Monensin 100 μM), and degranulation was assessed by evaluation of the lysosomal marker CD107a expression on CD56$^+$ CD3$^-$ cells by FACS analysis. To determine intracellular IFNγ production, cells were maintained in culture for 6 h in the presence of Brefeldine A (10 μg/ml), stained with anti-CD56 and -CD3 and subsequently fixed and permeabilized using Cytofix/cytoperm kit (BD Biosciences). After permeabilization, cells were stained with anti-IFN-γ specific mAb and analyzed by FACS.

**Mice treatment**. Starting at 8 weeks of age, male and female hSOD1$^{G93A}$, TDP43$^{A315T}$ or non-transgenic C57BL/6 J mice were randomly grouped for the treatments. NK cell depletion was performed using a blocking Ab against NK1.1, which recognizes an epitope of the NKR1Pc-activating receptor (PK136). Mice were i.p. injected with 200 μg (in 100 μl) of anti-NK1.1 Ab every 2 days the first week, every 4 days the second week and then repeated once a week until the age described in the text or until sacrifice for the survival analysis experiments. NK cell depletion from the blood sample was monitored by FACS[16,23]. For Ab anti-IFNγ treatment, mice were treated with 200 μg of rat XMG1.2, or control Ab isotype, by i.p. injection repeated every 5 days until the mice were sacrificed. For Ab anti-CCL2 administration, mice were i.p. treated with 200μg, every 4 days until the age described in the text or until sacrifice. Control mice were treated with the corresponding control IgG.

**Survival analysis**. hSOD1$^{G93A}$ and TDP43$^{A315T}$ mice were treated with Ab anti-NK1.1, XMG1.2 or vehicle as described above and were monitored daily. The endpoint was fixed when animals were unable to stand up within 20 s after being placed on either side, or death. The probability of survival was calculated using the Kaplan–Meier method, and statistical analyses were performed using a log-rank test.

**Behavioural tests**. Behavioural tests started when mice were 8-week-old. All animals were handled for at least 5 min/day for 2–3 days before starting the experiments. For the hindlimb extension reflex, mice were suspended by the tail, and scored for hindlimb extension reflex deficits. The scores were recorded from 0 to 2 as follows: 2, normal extension reflex in both hind limbs; 1.5, imbalanced extension in the hind limbs; 1.0, extension reflex in only one hindlimb; 0.5, the absence of any hindlimb extension; and 0, total paralysis. For the inverted grid test mice were placed in the center of a wire grid (40 × 60 cm, suspended 50 cm above a

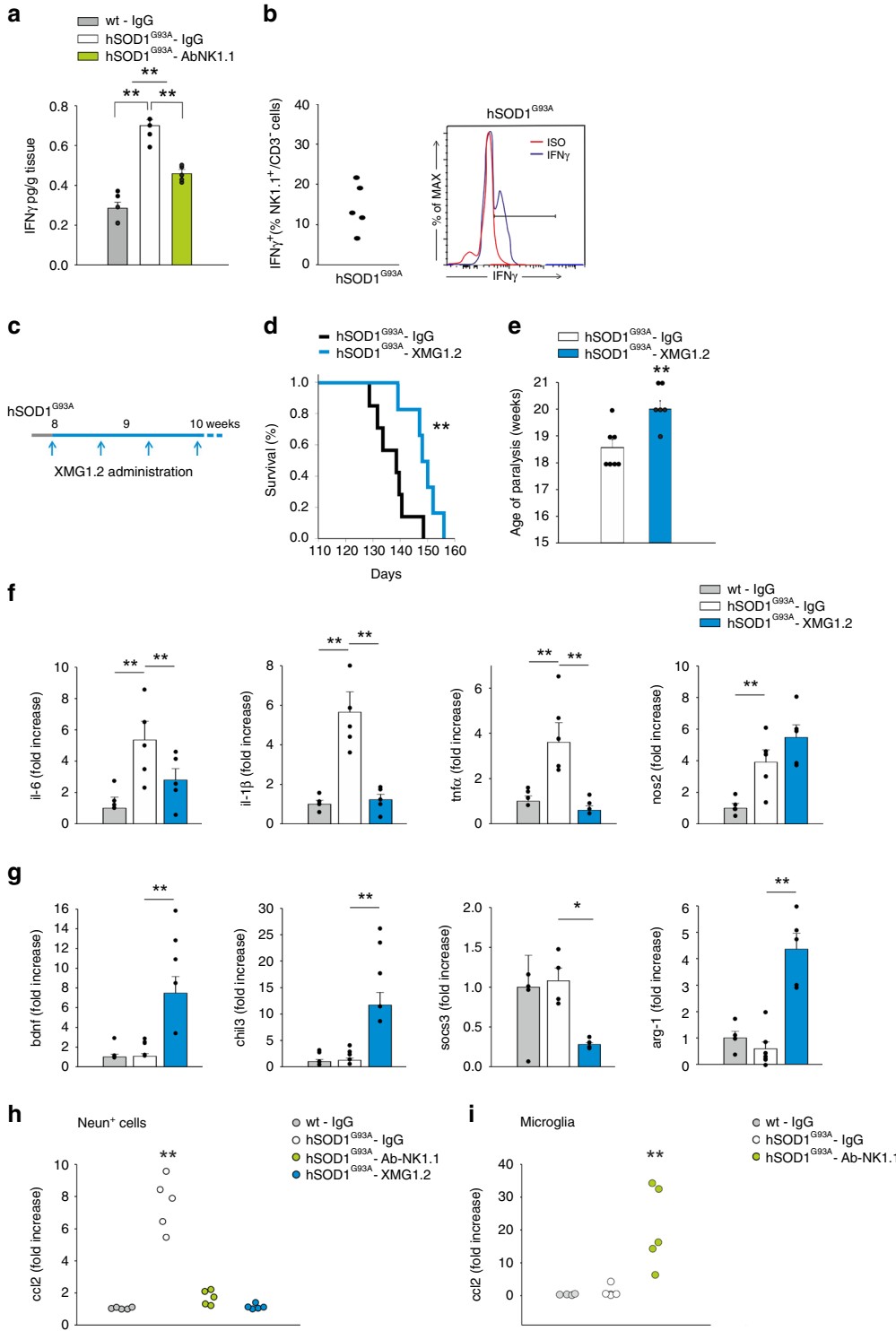

cushioned table) and then the grid was inverted (maximum time allowed 60 s). The time spent hanging on to the grid was measured. For the Rotarod test, motor coordination, strength and balance were assessed using a rotarod apparatus (Ugo Basile, Gemonio Italy, #47650). Animals were placed onto the cylinder at a constant speed of 15 rpm. The arbitrary cut-off time was 300 s., and the longest latency was recorded. For the hanging wire test mice were allowed to grab a horizontal wire with their front paws and the time spent hanging measured (maximum time allowed 60 s). Behaviour was scored according to the following scale: 1, hanging onto the bar with both forepaws; 2, in addition to 1, attempted to climb onto the bar; 3, hanging onto the bar with two forepaws and one or both hindpaws; 4, hanging onto the bar with all four paws with tail wrapped around the bar; 5, able to walk on the bar to escape.

**Immunostaining**. In different stages of pathology, mice were overdosed with chloral hydrate (400 mg/kg, i.p.) and then intra-cardially perfused with PBS and then PFA 4%; brains and spinal cord were then isolated, fixed in 4% formaldehyde and snap frozen. Cryostat sections (20 μm) were washed in PBS, blocked (3% goat serum in 0.3% Triton X-100) for 1 h, at RT, and incubated overnight at 4 °C with specific antibodies diluted in PBS containing 1% goat serum and 0.1% Triton X-100. The sections were incubated with the following primary Abs: anti-NKp46 (1:50), anti-CD31 (1:200), anti-Smi32 (1:500), anti-ChAT (1:100), anti-Mult-1 (1:100), anti-Foxp3 (1:100) and anti-Iba1 (1:500). For immunostaining on paraffin-embedded human motor cortex section, tissues were placed at 60 °C for 15 min, incubated in xylene at RT for 25 min, and then transferred sequentially into 100% EtOH, 95% EtOH, 70% EtOH, and 50% EtOH for 4 min at RT. The sections were

**Fig. 5 IFNγ mediates the effects of NK cells in ALS. a** Expression of IFNγ in the spinal cord of wt or hSOD1$^{G93A}$ mice (13 weeks) treated with control IgG or Ab-NK1.1 ($n = 5$, \*\*$P < 0.01$ power 1 one-way ANOVA). Error bars show mean ± SEM. **b** Frequency of IFNγ$^+$/NK1.1$^+$/CD3$^-$ cell population isolated from the spinal cord of hSOD1$^{G93A}$ mice ($n = 5$). Representative FACS analysis is shown on the right. **c** Scheme of mice treatment with XMG1.2. **d** Kaplan–Meier curve of hSOD1$^{G93A}$ mice treated with control IgG or XMG1.2 (hSOD1$^{G93A}$ $n = 7$ vehicle: 138.7 ± 2.5 days; $n = 6$ XMG1.2: 149.7 ± 2.3 days, data are the mean ± s.e.m. \*\*$P < 0.01$ two-sided log-rank test). **e** Age of paralysis, considered as the absence of hindlimb movement, upon XMG1.2 treatment, in hSOD1$^{G93A}$ mice ($n = 7$ vehicle: 18.6 ± 0.3 weeks; $n = 6$ XMG1.2: 20.0 ± 0.3 weeks. Data are the mean ± s.e.m. \*\*$P < 0.01$ power 0.80 one-way ANOVA). RT-PCR analysis of *il-6, il-1β, tnfα, nos2* (**f**); and *bdnf, chil3, socs3, arg-1* (**g**) gene expression in microglial cells isolated from the lumbar spinal cord of wt or hSOD1$^{G93A}$ mice (13 weeks) treated with control IgG or XMG1.2 ($n = 5$. Data are the mean ± s.e.m. wt vs hSOD1$^{G93A}$ and hSOD1$^{G93A}$ vs hSOD1$^{G93A}$ – XMG1.2. \*$P < 0.05$ \*\*$P < 0.01$ power >0.9 one-way ANOVA). **h** RT-PCR of *ccl2* expression in NeuN$^+$ cells isolated from the lumbar spinal cord of wt and hSOD1$^{G93A}$ mice (13 weeks) treated with control IgG, Ab-NK1.1 or XMG1.2 ($n = 5$, \*\*$P < 0.01$ power 0.859 vs wt mice, one-way ANOVA). **i** RT-PCR of *ccl2* expression in microglia isolated from the lumbar spinal cord of wt and hSOD1$^{G93A}$ mice (13 weeks) treated with control IgG or Ab-NK1.1 ($n = 5$, \*\*$P < 0.01$ power 0.859 vs wt mice, one-way ANOVA). Each dot (**h**, **i**) represents the qPCR analysis of ccl2 per mice. Source data are provided as a Source Data file.

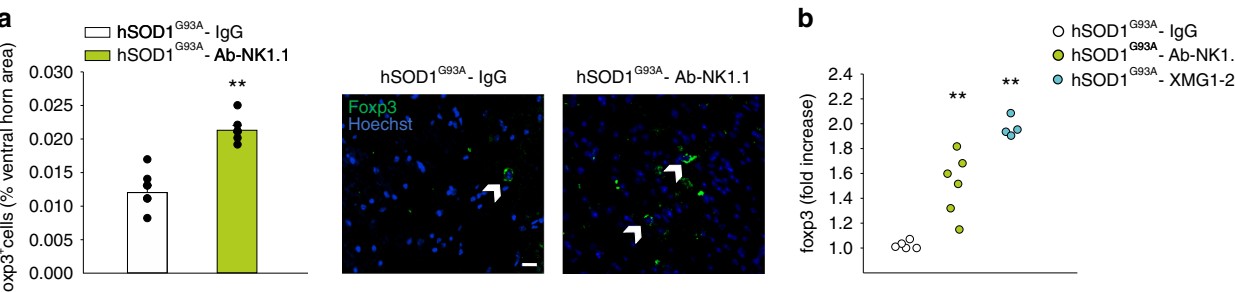

**Fig. 6 NK cells mediate Treg recruitment in the spinal cord of hSOD1$^{G93A}$ mice. a** Analysis of Foxp3$^+$ cells in the spinal cord of hSOD1$^{G93A}$ mice (13 weeks) treated with vehicle or Ab-NK1.1, expressed as % of spinal cord area ± s.e.m. ($n = 5$, \*\*$P < 0.01$ power 1 two-tailed Student's *t*-test). Representative immunofluorescence on the right. Error bars show mean ± SEM. Scale bar: 20 μm. **b** RT-PCR analysis of *foxp3* expression in the lumbar spinal cord of hSOD1$^{G93A}$ mice (13 weeks) treated with control IgG ($n = 5$), Ab-NK1.1 ($n = 6$) or XMG1.2 ($n = 5$), (\*\*$P < 0.01$ power 921 *vs* control, one-way ANOVA). Source data are provided as a Source Data file.

rinsed in deionized water and incubated with Abs: anti-Smi32 (1:500), anti-NKp46 (1:50) and anti-ULBP3 (1:100). After several washes, sections were stained with the fluorophore-conjugated antibody and Hoechst for nuclei visualization and analysed using a fluorescence microscope. For co-immunofluorescence, the secondary antibody was subsequently used. For Iba1 and Smi32 staining, coronal sections were first boiled for 20 min in citrate buffer (pH 6.0) at 95–100 °C.

**Motor neuron survival evaluation.** For MN survival, the whole ventral horns of lumbar spinal cord were photographed at ×20 magnification and digitized using a CoolSNAP camera (Photometrics) coupled to an ECLIPSE Ti-S microscope (Nikon) and processed using MetaMorph 7.6.5.0 image analysis software (Molecular Device). The number of MNs was evaluated counting only Smi32-positive cells with typical morphology triangular shape, single well-defined axon, large body diameter (≥20 μm) and intact axons and dendrites. This analysis was done in at least 12 serial slices for each animal.

**Isolation of NeuN-positive cells and extraction of total RNA.** The isolated spinal cords were cut in small pieces and single-cell suspension was achieved by enzymatic digestion in trypsin (0.25 mg/ml) solution in HBSS. The tissue was further mechanically dissociated using a wide-tipped glass pipette and the suspension applied to a 70 μm nylon cell strainer. Cells, obtained after a three-step Percoll gradient[45,46] were stained with anti-NeuN Ab (1:1000) at 4 °C for 30 min and isolated using a FACSAria II (BD Biosciences). Cell purity was at least 94%, as verified by flow cytometry and PCR analysis[47]. After cell sorting, total RNA was isolated by RNeasy Mini Kit and processed for real-time PCR.

**Isolation of lumbar microglia cells.** Adult microglia were isolated from the lumbar spinal cord tract of age-matched non-transgenic C57BL/6 J wt mice and hSOD1$^{G93A}$ mice[48]. In detail, mice were deeply anesthetized with chloral hydrate (i.p., 400 mg/Kg) before being transcardially perfused with PBS. Spinal cords were then flushed out from the spinal canal using a 20 ml syringe filled with PBS and digested with 30 units of papain (15–23 U/mg protein) for 30 min at 37 °C. Tissue was then triturated with a pipette to obtain single-cell suspensions, which were applied to 70 μm/40 μm cell strainers and used for the experiments. Purity of isolated microglia cells ranges between 70 and 90%, and were selected only the samples in which the purity was 90%.

**Real-time PCR.** Microglial cells and NeuN$^+$ cells sorted from spinal cord of mice were lysed in Trizol reagent for isolation of RNA. Reverse transcription reaction was performed in a thermocycler (MJ Mini Personal Thermal Cycler; Biorad) using IScript TM Reverse Transcription Supermix (Biorad) according to the manufacturer's protocol, under the following conditions: incubation at 25 °C for 5 min, reverse transcription at 42 °C for 30 min, inactivation at 85 °C for 5 min. Real-time PCR (RT-PCR) was carried out in a I-Cycler IQ Multicolor RT-PCR Detection System (Biorad) using SsoFast EvaGreen Supermix (Biorad) according to the manufacturer's instructions. The PCR protocol consisted of 40 cycles of denaturation at 95 °C for 30 s and annealing/extension at 60 °C for 30 s. For quantification analysis, the comparative Threshold Cycle (Ct) method was used. The Ct values from each gene were normalized to the Ct value of GAPDH in the same RNA samples. Relative quantification was performed using the 2ΔDDCt method[49] and expressed as fold change in arbitrary values. The primers used were listed in Supplementary Table 1.

**Cytotoxicity and degranulation assays.** Cell viability of spinal cord primary culture was determined by MTT assay or by staining dead cells[50]. Results are expressed as percentage of cell survival, taking as 100% the cells not incubated with NK cells. In order to analyse NK cell degranulation, purified NK cells were activated O/N in IL-15 (50 ng/ml) and then co-incubated with MNs or GL261 for 4 h at the Effector:Target (E:T) ratios indicated in Fig. 3d and Supplementary Figs. 3 and 4. The last two hours, FITC-conjugated anti-mouse CD107a or IgG were added directly into each well. NK cells were collected from the non-adherent fraction and analysed by FACS. Cell viability of spinal cord primary culture was determined by MTT assay or by staining dead cells. Results are expressed as percentage of cell survival, taking as 100% the cells not incubated with NK cells.

**Morphological analysis of Iba1$^+$ cells and 3D reconstruction.** Brain and lumbar spinal cord slices from perfused mice were analysed by confocal microscopy and skeleton analysis to assess Iba1$^+$ cell morphology. Twenty mm z-stacks were acquired at 0.5 mm intervals using an FV1000 laser scanning microscope (Olympus) at ×60 objective. Maximal intensity projections of each image were generated, binarized, and skeletonized using the Skeletonize 2D/3D plugin in ImageJ, after which the Analyze Skeleton plugin (http://imagej.net/AnalyzeSkeleton) was applied. The average branch number (process end points per cell) and length per cell were recorded for each image with a voxel size exclusion limit of 150 applied. The number of single and multiple junction points was additionally

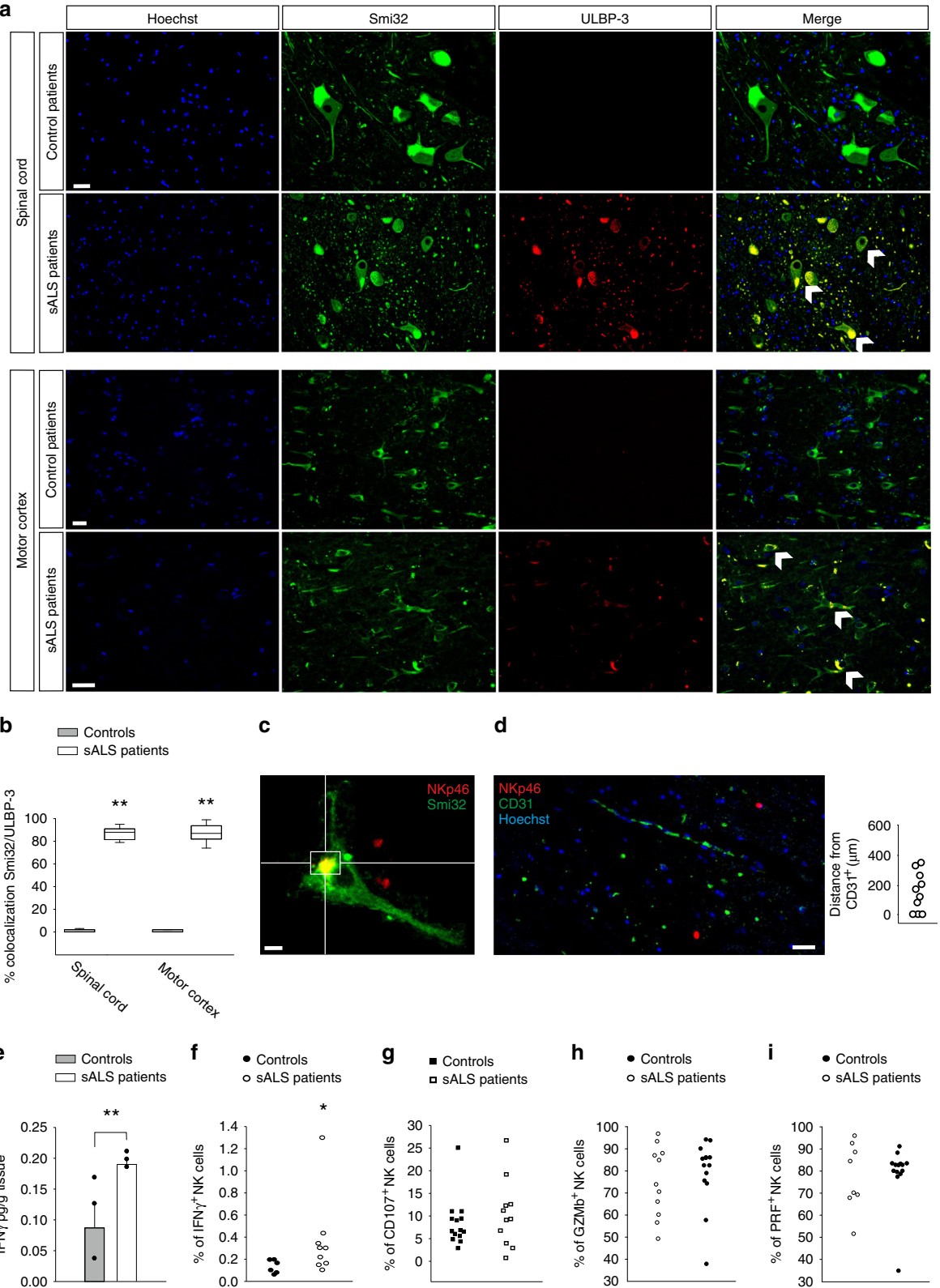

calculated to give an indication of branching complexity. The areas of the soma and scanning domain were measured for each cell. 3D reconstruction of NK cells was achieved by confocal microscopy analysis with a FV1200 (Olympus) Laser Scanning System, at ×60 magnification. NK cells were identified with an Alexa fluor 594 conjugated NKp46 antibody. Acquisition files were then processed with ImageJ software for two-dimensional analysis. Three-dimensional reconstructions were generated with Imaris software (Bitplane, Zurich, Switzerland).

**Image acquisition and data analysis**. Images were digitized using a CoolSNAP camera (Photometrics) coupled to an ECLIPSE Ti-S microscope (Nikon) and processed using MetaMorph 7.6.5.0 image analysis software (Molecular Device).

Slices were scanned by consecutive fields of vision (×10 objective lens) to build a single image per section. Data were expressed as area occupied by fluorescent cells versus total area by converting pixel to mm. For comparison between different treatments, at least 12 coronal sections per lumbar spinal cord or brain were analysed.

**Fig. 7 NK cell analyses in sALS patients. a** Expression of ULBP-3 in the spinal cord and cerebral motor cortex of control or sALS patients, showing co-staining with Smi32$^+$ cells ($n = 8$ controls, 12 sALS patients). Scale bar: 20 μm. **b** Colocalization of Smi32 and ULBP-3 covered area in the spinal cord and cerebral motor cortex of control or sALS patients ($n = 8$ controls, 12 sALS patients **$P < 0.01$ power 1 two-tailed Student's $t$-test). **c** Representative image of NK cell–MN contacts in the spinal cord of ALS patient (original magnification, x600). Scale bar: 10 μm. ($n = 8$ ALS patients). For boxplots, the center line, boxes and whiskers represent the median, inner quartiles, and rest of the data distribution, respectively. **d** Distances of NKp46$^+$ cells from vessel endothelial cells (CD31$^+$ cells) in the spinal cord of ALS patient ($n = 12$ sALS patients). Scale bar: 20 μm. Representative immunofluorescence is shown. **e** Expression of IFNγ in the spinal cord of control and sALS patients ($n = 3$, **$P < 0.01$ power 0.871 two-tailed Student's $t$-test). Error bars show mean ± SEM. **f** Frequency of IFNγ$^+$ cells in the CD56$^+$/CD3$^-$ cell populations (NK cells) isolated from the peripheral blood of control or sALS patients ($n = 6$ controls, 9 sALS patients. *$P < 0.05$ two-tailed Student's $t$-test). Note that when the highest value among sALS patients was excluded, the difference among the two groups remains significant ($p < 0.034$ power 0.716). **g** NK cells, isolated from the peripheral blood of control or sALS patients, were incubated with human K562 cells. Degranulation was assessed by FACS analysis of CD107$^+$ cells (%, minus degranulation in the absence of targets) ($n = 14$ controls, 11 sALS patients). **h, i** Frequency of GZMb$^+$ (**h**) and PRF$^+$ (**i**) cells in the CD56$^+$/CD3$^-$ cell populations (NK cells) isolated from the peripheral blood of control or sALS patients ($n = 8$ controls, 15 sALS patients). Source data are provided as a Source Data file.

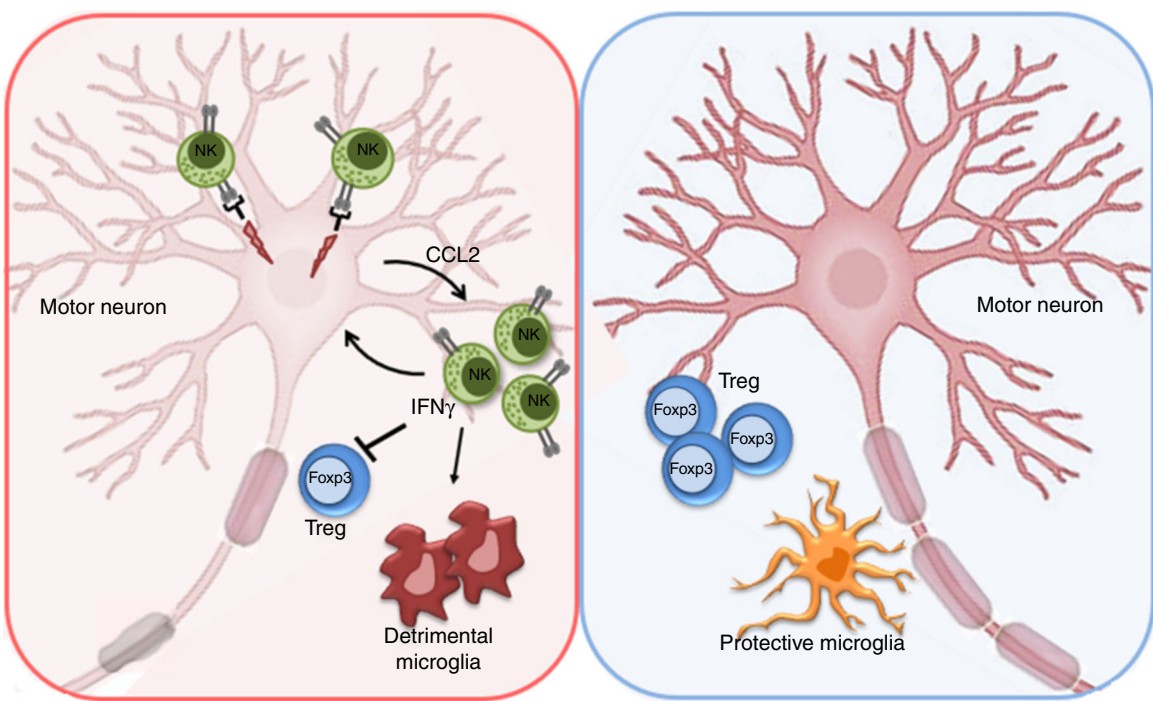

**Fig. 8 Graphic summary.** (Left) NK cells' effects in the spinal cord of hSOD1$^{G93A}$ mice at the early symptomatic stage. NK cells directly exert cytotoxic activity against MNs through the NKG2D-Mult-1 axis. Furthermore, IFNγ released by NK cells: (i) increases the release of CCL2 by damaged neurons; (ii) reduces the number of Treg Foxp3$^+$ cells; (iii) shapes microglia toward a pro-inflammatory phenotype. (Right) NK cell depletion, in hSOD1$^{G93A}$ mice, induces neuroprotective microglia phenotype, increases the number of Treg Foxp3$^+$ cells, protecting the MNs.

**Measurement of IFNγ by ELISA**. The spinal cord of control and sALS patients, or 13-weeks-wt and hSOD1$^{G93A}$ mice were disrupted with a homogenizer and analysed for IFNγ content using a sandwich ELISA, following the manufacturer's instructions. Briefly, 96-well ELISA microplates were coated with anti- IFNγ monoclonal Ab. Samples or IFNγ standard were added at the appropriate dilution and incubated for 2 h at room temperature. After careful washing, biotinylated goat anti-human IFNγ was added to each well; horseradish-peroxidase was used as secondary Ab and optical density was read at 450 nm.

**Statistical analysis**. Data are shown as the mean ± SEM. Statistical significance was assessed by Student's $t$-test, one-way ANOVA or two-way ANOVA for parametrical data, as indicated; Holm–Sidak test was used as a post-hoc test; Mann–Whitney Rank test and Kruskal–Wallis for non-parametrical data, followed by Dunn's or Tukey's post-hoc tests. For multiple comparisons, multiplicity-adjusted p-values are indicated in the corresponding figures (*$p < 0.05$, **$p < 0.01$). Power calculation (power) was added in all the figure legends. For the Kaplan–Meier analysis of survival, the log-rank test was used. Statistical analyses comprising calculation of degrees of freedom were done using Sigma Plot 11.0, Imaris; Origin 7, and Prism 7 software.

**Reporting summary**. Further information on research design is available in the Nature Research Reporting Summary linked to this article.

## Data availability
The data that support the findings of this study are available from the corresponding author upon reasonable request. The source data are provided as a Source Data file.

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

## Acknowledgements

S.G. is supported by FIRC-AIRC fellowship for Italy 22329/2018 and by Pilot ARISLA NKINALS 2019. C.L. is supported by MIUR PRIN 2015, by MoH and by AIRC 2015 IG16699. L.P., A.S. and C.L. are supported by Euronanomed II (H2020); E.A. by ALS Stichting grant The Dutch ALS Tissue Bank. G.C. was a former PhD student in the Center for Life Nano Science at IIT, Rome. We acknowledge the team who helped in the collection of ALS tissue samples (Prof. dr. D. Troost, Prof. dr. M. de Visser, Dr. A.J. van der Kooi and Dr. J. Raaphorst, and E. Onesti).

## Author contributions

S.G.: performed and ideated most of the experimental work and wrote the paper; G.C.: contributed to many experimental activities for mice manipulation and behavioural tests; A.P.: performed FACS analyses; M.I.: provided patient's blood and clinical information; M.R. and F.S.: produced and analysed transgenic mice phenotype; E.A.: produced and provided human CNS tissues from patients and clinical information; G.B.: performed FACS analyses and immune cell activity assays and wrote the paper; L.P.: synthesized the dendrimers and supervised the experiments with dendrimers and wrote the paper; R.M.R.: supervised experiments on microglia gene expression and wrote the paper; A.S.: supervised experiments on NK cells and wrote the paper; C.L.: supervised all the experimental work and wrote the manuscript.

## Competing interests

The authors declare no competing interests.
