## [Peer Review File · Nature Communications]

Reviewers' comments:

Reviewer #1 (Remarks to the Author):

Garofalo and colleagues investigate the functional impact of NK cells on disease onset and progression in mouse models of amyotrophic lateral sclerosis (ALS). Non-cell autonomous effects of glial cells and immune cells, such as circulating CD4+ and CD8+ T-cells, on motor neuron degeneration have been previously shown in ALS models, however a potential role for NK cells in ALS has yet to be demonstrated, until now in the present manuscript.

The authors present ALS patient and mouse data showing infiltration of NK cells into affected CNS regions which occurs early in SOD1 mice. Using a series of neutralising antibody experiments in vitro and in vivo, they provide evidence that depletion of NK cells or NK cell ligands expressed by motor neurons extends survival in both SOD1 and TDP-43 mouse models, without affecting onset or progression. This effect was attributed to an IFN γ -dependent microglial and Treg mechanisms, again supported by IFN γ neutralising antibody treatment experiments in mutant SOD1 mice. These findings uncover a new potential immune cell player in ALS, NK cells, which may influence microglial-mediated neuroinflammation, Treg infiltration, neurodegeneration and outcome in ALS.

Overall, this is a comprehensive study yielding new insights into non-cell autonomous influences on motor neuron degeneration in ALS. The strengths of the study are the use of human tissue and fluid samples, two leading mouse models of ALS, NK cell and motor neuron co-cultures and a series of well designed neutralising antibody treatment experiments to unravel the functional impacts of NK cells and their putative cytokine mediator in ALS. The manuscript could be strengthened by addressing the following:

1. Modulation of NK cells in mutant SOD1 mice is not a novel concept. Komine and co-workers showed that NK1.1 antibody treatment did not affect survival in mutant SOD1 mice (Komine et al. 2018 CDD 25:2130-2146). Please cite this paper and account for this important discrepancy?
2. Modulation of microglia cells and Tregs affects rate of disease progression in mutant SOD1 mice, whereas this study shows NK cell depletion lengthens survival without affecting progression. If NK cells are working through microglia and Tregs, how do the authors explain this difference?
3. The authors need to explain the relevance and significance of some experiments e.g. CD56^{high} vs. CD56^{low} NK cells in Fig 1. Why measure both? Also, CD19+ cells in Suppl Fig 1. Lastly, CD107a+ degranulating NK cells.
4. The human group sizes in Fig 1a are surprisingly small (n=3) for this level of publication. Fig 1a and 6a lack quantification to demonstrate variability in ALS patients, even if controls have negligible signal.
5. The authors need to provide evidence that their NK cell depletion strategy in Fig 1 successively reducing NK cell levels.
6. In Fig 2e, the authors should have used a scrambled siRNA control, not empty treatment, to control for sequence-independent effects.
7. In all experiments using an isotype antibody control, please state this on figure labels, instead of just hSOD1G93A e.g. Fig. 2c.
8. Does NK1.1 antibody treatment affect hSOD1 expression level, like mSOD1, in Fig 3c?
9. The SALS group appears only significant due to one patient in Fig 6c. Is this an outlier?

10. In Suppl Fig 1, there is not apparent accumulation of CD19+ or CD4+ cells, as stated on p.4.
11. Can the authors show hanging wire data for male mice in Suppl Fig 2?
12. The male and female healthy controls don't add up to 14 in Table 2.
13. Can the authors confirm purity of NeuN+ cells in Fig 2?
14. There is a paragraph duplication on p.6 in results.
15. The authors state hSOD1G93A NK cells have similar cytotoxic activity to WT, but this is not shown in Suppl Fig 3?
16. Please explain the significance of the findings that rae-1 and pvr are not overexpressed in NeuN+ cells in Fig 2b.
17. Please change first line of discussion to "We identified a new potential mechanism to explain the non-cell autonomous motor neuron degeneration..."

Reviewer #2 (Remarks to the Author):

This is an important paper and is, with the right revisions, a good candidate for being published in Nature Communications. It essentially builds on a Cell paper published in January 2019, which describes an endogenous ligand for the natural killer (NK) cell receptor NKG2D, Retinoic Acid Early 1 (RAE1), is re-expressed in adult dorsal root ganglion neurons following peripheral nerve injury, triggering selective degeneration of injured axons by Infiltration of cytotoxic NK cells. The current study shows that this pathway is somewhat generalizable (to animal models of ALS and also provides additional circumstantial evidence in tissue from post-mortem cases). I would like to see the following addressed:

- 1) The abstract starts with: "Motor neuron degeneration in amyotrophic lateral sclerosis (ALS) reflects their abnormal interactions with immune cells and glia." This may be misleading as it suggests that motor neuron death would not occur if immune cells and glia were not in the context, which I do not believe is the case (and has not been definitively shown). It would be more accurate to state that immune cells and glia have clearly been shown to contribute to motor neuron death.
- 2) A mouse model of familial ALS, hSOD1G93A was used in this study. Can the authors confirm the number of copies /degree of overexpression and whether a wildtype overexpressor was used as a control?
- 3) Can the authors comment on how closely the TDP43 A315T reflects the molecular and cellular pathophysiology and clinical phenotype?
- 4) With the NK1.1 antibody therapy, how did this affect the levels of other immune cells? NK cell depletion enhancing survival could be direct or indirect (i.e. through the effects of NK cell depletion on other cell types). Could the authors comment on this?
- 5) Can different states of NK cell activation be reproducibly identified?
- 6) As above, Davies et al 2019 Cell already shows that an endogenous ligand for the natural killer (NK) cell receptor NKG2D, Retinoic Acid Early 1 (RAE1), is re-expressed in adult dorsal root ganglion neurons following peripheral nerve injury, triggering selective degeneration of injured axons and that Infiltration of cytotoxic NK cells into the sciatic nerve by extravasation occurs

within 3 days following crush injury. Therefore, although this work bears significance in ALS and does show some evidence of this phenomenon in human tissue, it remains somewhat incremental. What would be desirable is further mechanistic insight at a molecular level into i) how the NK cells are 'activated' in such circumstances and ii) how motor neurons actually die as a result of this ligand – receptor interaction? Is this just by secretory lysosome exocytosis? Can the authors show this more clearly? Perhaps by immunocytochemistry against key stage specific proteins across the 4 stages of this process? If the authors in ex vivo / in vitro models perturbed NK cell exocytic machinery by knock down for example of WASP, the AP-3 complex, Rab27a, Munc13-4 or syntaxin 11, does this abrogate toxicity? Might alternative pathways be active here?

7) Is this also relevant to other neurodegenerative conditions such as Alzheimers and PD?

8) The study, whilst convincing, tends to rely on antibody mediated perturbations. I feel that orthogonal validation using e.g. genetic ablation studies / mice carrying a deletable mutant gene driven off a cell type-specific promoter would also add weight to findings

9) An additional marker beyond SMI32 should be used to identify motor neurons in the assays evaluating effects on NK depletion

10) Can the authors include post-mortem delays for each of the cases

11) Can power calculations please be shown

12) Can an ethics statement please be provided

Reviewer #3 (Remarks to the Author):

Critique of "Natural killer cells modulate motor neuron-immune cell cross talk in Amyotrophic Lateral Sclerosis"

1. the figures are not numbered and the subfigures do not have heading, which makes the review difficult. Notwithstanding, the study is very carefully done and confirms nicely the hypothesis of role of NK cells in animal model using Ab-CCL2 and NK cell depletion.

2. The role of NK cells is important, but it needs to be demonstrated in ALS patients' spinal cord by visualization of NK cells. The figure shows NK cells in cortex but there is no relation to neurons or vessels. A description of localization is needed. Localization in anterior horns in vicinity of neurons could firm up the role of NK cells in ALS patients

3. The title should everywhere specify "ALS model" not "ALS". This study points to the role of NK cells in animal models but has no neuropathological demonstration of NK cells in ALS patients' spinal cord, although it has strong evidence in animal model

4. This study has strong point of colocalization of SMI32 with Mult1 and other interactions, expression of IFN gamma with NK cells, etc .

5. The study overlooks the likely pathogenic role of cytotoxic CD8 T cells expressing granzyme and perforin in the spinal cord anterior horns of ALS patients (1, 2).

6. This sentence is incomplete: Since NKG2D ligands are involved in triggering the cytotoxicity of NKG2D receptor-expressing NK cells

7. Human data in Supl fig 1 just show ctrl vs. patients but there is no time of blood collection vs. onset of disease, which is critical to understand role in disease.

Overall, this is a very strong work but it is difficult to review by non-afficionado of mouse models and mouse NK cells. It would be much easier to review if the legends were appended to pictures and each subfigure had a heading.

1. Fiala M, et al. (2010) IL-17A is increased in the serum and in spinal cord CD8 and mast cells of ALS patients. *Journal of neuroinflammation* 7(1):76-90.
2. Lam L, et al. (2016) Epigenetic changes in T-cell and monocyte signatures and production of neurotoxic cytokines in ALS patients. *FASEB journal : official publication of the Federation of American Societies for Experimental Biology* 30(10):3461-3473.

Reviewer #4 (Remarks to the Author):

In the current manuscript the authors unravel a yet unknown role of NK cells in the onset of motor neuron degeneration in ALS and their impact on Treg recruitment and microglia. Combining research on human biomaterial with ALS mouse models the authors demonstrate that NK cells infiltrate the motor cortex and spinal cord most likely in a CCL2-dependent manner. NK-cell mediated cytotoxicity against motor neurons is driven by NKG2D expression. Accordingly, NK cell depletion reduced the pace of motor neuron degeneration, delayed the onset of motor impairment and increased survival in the mouse model. Finally, the authors also showed that IFN- γ produced by NK cells shifts microglia towards an inflammatory phenotype and reduces infiltration of Treg cells.

Although this study is well-designed, the following points need to be addressed.

Major points:

- The mechanism of NK-mediated motor neuron cytotoxic activity needs to be further elucidated. The authors should show, both in the human and mouse tissue, whether NK cells in contact with motor neurons express granzymes and/or perforin. Furthermore, the impact of granzyme and/or perforin blockade should be assessed in the mouse model.
- Although the authors demonstrated that NK cells isolated from the peripheral blood of sALS patients show no enhanced cytotoxic capacity towards K562 target cells when compared to healthy individuals, it would be interesting to see whether granzyme and perforin levels differ between these two groups.
- Are there any evidences, e.g. genetic, that NK cells are involved in the pathogenesis of ALS?
- The authors describe diminished NK-cell proportions in the periphery of ALS patients. How do they explain this finding in the context that another study found increased NK cell numbers (reference 18 in the manuscript)?
- What was the impact of CCL2 blocking on other immune subsets; particular CD8 T cells.
- Along this line, in addition to NK cells T cells can also produce IFN- γ . The authors should take this into consideration.

Minor points:

- For a broader readership the mouse models should be better explained in the results part.
- In the first paragraph of page 8 the authors should already explain which ligand (Mult-1, Nectin-2, Rae-1, Pvr) belongs to which receptor (NKG2D, DNAM-1).
- N-number for the human tissues should be already provided in the result part or respective figure legend.

To the Reviewers:

We thank the reviewers for their general appreciation of the significance and novelty of the research, and for their constructive suggestions. Here is the answer to their comments.

Reviewers' comments:

Reviewer #1 (Remarks to the Author):

Garofalo and colleagues investigate the functional impact of NK cells on disease onset and progression in mouse models of amyotrophic lateral sclerosis (ALS). Non-cell autonomous effects of glial cells and immune cells, such as circulating CD4+ and CD8+ T-cells, on motor neuron degeneration have been previously shown in ALS models, however a potential role for NK cells in ALS has yet to be demonstrated, until now in the present manuscript.

The authors present ALS patient and mouse data showing infiltration of NK cells into affected CNS regions which occurs early in SOD1 mice. Using a series of neutralising antibody experiments in vitro and in vivo, they provide evidence that depletion of NK cells or NK cell ligands expressed by motor neurons extends survival in both SOD1 and TDP-43 mouse models, without affecting onset or progression. This effect was attributed to an IFN γ -dependent microglial and Treg mechanisms, again supported by IFN γ neutralising antibody treatment experiments in mutant SOD1 mice. These findings uncover a new potential immune cell player in ALS, NK cells, which may influence microglial-mediated neuroinflammation, Treg infiltration, neurodegeneration and outcome in ALS.

Overall, this is a comprehensive study yielding new insights into non-cell autonomous influences on motor neuron degeneration in ALS. The strengths of the study are the use of human tissue and fluid samples, two leading mouse models of ALS, NK cell and motor neuron co-cultures and a series of well designed neutralising antibody treatment experiments to unravel the functional impacts of NK cells and their putative cytokine mediator in ALS. The manuscript could be strengthened by addressing the following:

1. Modulation of NK cells in mutant SOD1 mice is not a novel concept. Komine and co-workers showed that NK1.1 antibody treatment did not affect survival in mutant SOD1 mice (Komine et al. 2018 CDD 25:2130-2146). Please cite this paper and account for this important discrepancy?

RESPONSE We thank the reviewer for this observation that affords us the opportunity to clarify that there are no discrepancies with these results: in the paper of Komine et al., 2018, the authors depleted NK cells from mice starting from 14 weeks of age. At that age, we observed that NK cells already infiltrated the CNS of hSODG93A mice, the microenvironment is modified and motor neurons are affected. In our conditions, we depleted NK cells starting from an earlier stage (8 weeks), thus preventing NK cells entry in the CNS. This is confirmed by our experiments when, starting treatment from 13 weeks of age, we did not detect effects on survival time (data previously not inserted, now mentioned in the results together with the new reference: “The timing of NK cell depletion is crucial, because when NK1.1 treatment of hSOD1^{G93A} mice started at 13 weeks of age, no effects on survival time was observed (data not shown), in accordance with another study²⁴” Page: 6 lines 4-6).

2. Modulation of microglia cells and Tregs affects rate of disease progression in mutant SOD1 mice,

whereas this study shows NK cell depletion lengthens survival without affecting progression. If NK cells are working through microglia and Tregs, how do the authors explain this difference?

RESPONSE Again, we thank the reviewer for raising this point for discussion. We show that NK cells may both have a direct neurotoxic effect on motor neurons and an indirect deleterious effect, modulating microglia and Treg activity in SOD1 mice. Since the direct effects on motor neurons affect the local microenvironment, shifting the balance of protective and detrimental signals, our hypothesis is that different results obtained by a direct alteration of microglia and Treg in SOD1 mut mice might be explained by the lack of direct effects on motor neurons and by a different efficacy and quality of microenvironment modulation. This is now considered in the Discussion “While Treg and microglia modulation correlates with disease progression in mutant SOD1 mice and ALS patients^{3,21,29,38}, we show that NK cell-mediated modulation of these cell types only affects survival and onset time, possibly due to specific or partial alteration of microglia and Treg phenotype in the different experimental settings, and to an additional direct neurotoxic effect of NK cells.” (Page 13: Lines 5-8:).

3. The authors need to explain the relevance and significance of some experiments e.g. CD56high vs. CD56low NK cells in Fig 1. Why measure both? Also, CD19+ cells in Suppl Fig 1. Lastly, CD107a+ degranulating NK cells.

RESPONSE: 1) The reason to measure both CD56 high and low was to possibly identify different functions for these NK cell subtypes in patients. 2) CD19⁺ B cells were measured to investigate whether we could detect B cell alteration in our cohort of ALS patients; 3) CD107a⁺ cells represent actively degranulating NK cells. We now better introduce all these experiments in the Results (pages 4-5 and 7).

4. The human group sizes in Fig 1a are surprisingly small (n=3) for this level of publication. Fig 1a and 6a lack quantification to demonstrate variability in ALS patients, even if controls have negligible signal.

RESPONSE We now increased the number of ALS patients (n=12) and controls (n=8) for both Fig.1a and Fig. 6a (now 7a, also for human spinal cord tissues). Quantification is now provided for both sets of results. A new Fig 7b shows statistics for co-localization studies (Smi32/ULBP-3).

5. The authors need to provide evidence that their NK cell depletion strategy in Fig 1 successively reducing NK cell levels.

RESPONSE The efficacy of this treatment was previously published (Garofalo et al. 2015: supplementary Fig 4c of ref 22). However, in addition to mention this paper, we now state that NK cell depletion was checked and confirmed in the current experiments (page 5 last line and first line page 6).

6. In Fig 2e, the authors should have used a scrambled siRNA control, not empty treatment, to control for sequence-independent effects.

RESPONSE We thank the reviewer for this important suggestion. We add in vitro and in vivo data of NKG2D reduction with Dendrimers loaded with the scrambled siRNA as control. These new data are now shown in the new figure 3f and Suppl. Figure 4 and methods modified accordingly.

7. In all experiments using an isotype antibody control, please state this on figure labels, instead of just hSOD1G93A e.g. Fig. 2c.

RESPONSE We now better specify the use of an isotype antibody control in Methods, paragraph “Mice treatment” and on all figure labels (IgG).

8. Does NK1.1 antibody treatment affect hSOD1 expression level, like mSOD1, in Fig 3c?

RESPONSE As suggested by the reviewer, we performed qPCR analysis for hSOD1 in NK cell depleted mice, and the results (BELOW) indicate that there is no variation (data are inserted in the results –page 9, line 10- as “not shown, n=5 mice per treatment”).

9. The SALS group appears only significant due to one patient in Fig 6c. Is this an outlier?

RESPONSE We thank the reviewer for this observation. We performed a statistical analysis excluding that patient, and the differences remain significant among the two groups ($p < 0.034$), now described in legend to figure 7.

10. In Suppl Fig 1, there is not apparent accumulation of CD19+ or CD4+ cells, as stated on p.4.

RESPONSE We apologize for the use of a unique y axis for such different numbers. We now report data to show the accumulation of CD19+ and CD4+ cells in SOD mutant mice, with respect to wt mice (dotted line), and we better describe in figure legend that all the infiltrated cell population in hSOD1 mice were significantly increased in comparison with wt mice. Statistical analysis is shown vs 10 weeks-old mut SOD mice. A new Suppl Fig 1 is provided.

11. Can the authors show hanging wire data for male mice in Suppl Fig 2?

RESPONSE Yes, we performed the hanging wire test for male SOD mice, and these results are now shown in Suppl Fig 2. Text is modified accordingly.

12. The male and female healthy controls don't add up to 14 in Table 2.

RESPONSE We apologize for the mistake. Table 2 has been modified according to the real total number of controls (14).

13. Can the authors confirm purity of NeuN+ cells in Fig 2?

RESPONSE The purity of NeuN⁺ cells is confirmed and described in the methods (page 21 lines 5-6).

14. There is a paragraph duplication on p.6 in results.

RESPONSE We thank the reviewer, the duplicated paragraph has been eliminated.

15. The authors state hSOD1G93A NK cells have similar cytotoxic activity to WT, but this is not shown in Suppl Fig 3?

RESPONSE We are sorry for the lack of clarity, and we now better explain the experiments shown in Suppl Fig.3, adding a label on top of figures (wt NK cells– fig 3b– and sodg93a NK cells - fig 3c-). Our statement refers to the effect shown in fig 3b (right) vs fig 3c. The effect of NK cell isolated from wt mice (Fig. 3b right) and incubated with wt or SOD MN is the same of that of NK cells isolated from hSODG93A mice (Fig. 3c) and incubated with wt or SOD MN. Furthermore, we now insert CD107a expression in hSOD NK cells incubated with wt or hSOD MN (NEW SUPPL. FIG. 3c, LEFT). We thank the reviewer to give us the opportunity to better explain our results.

16. Please explain the significance of the findings that *rae-1* and *pvr* are not overexpressed in NeuN⁺ cells in Fig 2b.

RESPONSE We have no experimental explanations for the lack of overexpression of *rae-1* and *pvr* in NeuN⁺ cells, but we hypothesize that this is part of the functional redundancy of these ligands (Lanier, 2005 Ann Rev Immunol) whose genes undergo different control of expression. We inserted a sentence in the results to consider this difference (Page 6, line 23): “This selective overexpression may reflect the functional redundancy of these ligands”.

17. Please change first line of discussion to "We identified a new potential mechanism to explain the non-cell autonomous motor neuron degeneration..."

RESPONSE Modified.

Reviewer #2 (Remarks to the Author):

This is an important paper and is, with the right revisions, a good candidate for being published in Nature Communications. It essentially builds on a Cell paper published in January 2019, which describes an endogenous ligand for the natural killer (NK) cell receptor NKG2D, Retinoic Acid Early 1 (RAE1), is re-expressed in adult dorsal root ganglion neurons following peripheral nerve injury, triggering selective degeneration of injured axons by Infiltration of cytotoxic NK cells. The current study shows that this pathway is somewhat generalizable (to animal models of ALS and also provides additional circumstantial evidence in tissue from post-mortem cases). I would like to see the following addressed:

1) *The abstract starts with: “Motor neuron degeneration in amyotrophic lateral sclerosis (ALS) reflects their abnormal interactions with immune cells and glia.” This may be misleading as it suggests that motor neuron death would not occur if immune cells and glia were not in the context, which I do not believe is the case (and has not been definitively shown). It would be more accurate to state that immune cells and glia have clearly been shown to contribute to motor neuron death.*

RESPONSE Following the reviewer suggestion, we modify the text, as “In amyotrophic lateral sclerosis (ALS), immune cells and glia contribute to motor neuron degeneration”.

2) *A mouse model of familial ALS, hSOD1G93A was used in this study. Can the authors confirm the number of copies /degree of overexpression and whether a wildtype overexpressor was used as a control?*

RESPONSE The transgene copy number is typically between 17-23. We did not use wild-type overexpressor as controls, but non tg-littermates. This information is in the methods (page 15 lines 21-22 page 16 lines 1-2).

3) *Can the authors comment on how closely the TDP43 A315T reflects the molecular and cellular pathophysiology and clinical phenotype?*

RESPONSE The TDP43A315T model is now introduced in the abstract and results (page 5 lines 20-23).

4) *With the NK1.1 antibody therapy, how did this affect the levels of other immune cells? NK cell depletion enhancing survival could be direct or indirect (i.e. through the effects of NK cell depletion on other cell types). Could the authors comment on this?*

RESPONSE We did investigate the effect of NK cell depletion on the infiltration level of other immune cells in the SC and MC of SOD mice (13 weeks old). The results show that NK cell depletion did not change the frequency of infiltrating CD8⁺ and CD4⁺ cells in the motor cortex and the spinal cord of mut SOD mice. This is now described and commented in the text (Page 6 Lines 11-13).

5) *Can different states of NK cell activation be reproducibly identified?*

RESPONSE As far as we know, there are no generally accepted ways to identify NK activation states. Thus, to confirm our previous observation obtained with the analysis of IFN- γ expression levels in NK cells from spinal cord, we analyzed the combined expression of CD69, GZMb and perforin in spinal cord NK cells from 13 and 16 weeks old SOD and wt mice and compared it with the expression in spleen. The results are shown in the new Fig. 2b,c and commented in the text (Page 4 Lines 20-25).

6) *As above, Davies et al 2019 Cell already shows that an endogenous ligand for the natural killer (NK) cell receptor NKG2D, Retinoic Acid Early 1 (RAE1), is re-expressed in adult dorsal root ganglion neurons following peripheral nerve injury, triggering selective degeneration of injured axons and that Infiltration of cytotoxic NK cells into the sciatic nerve by extravasation occurs within 3 days following crush injury. Therefore, although this work bears significance in ALS and does show some evidence of this phenomenon in human tissue, it remains somewhat incremental. What would be desirable is further mechanistic insight at a molecular level into i) how the NK cells are 'activated' in such circumstances and ii) how motor neurons actually die as a result of this ligand – receptor interaction? Is this just by secretory lysosome exocytosis? Can the authors show this more clearly? Perhaps by immunocytochemistry against key stage specific proteins across the 4 stages of this process? If the authors in ex vivo / in vitro models perturbed NK cell exocytic machinery by knock down for example of WASP, the AP-3 complex, Rab27a, Munc13-4 or syntaxin 11, does this abrogate toxicity? Might alternative pathways be active here?*

RESPONSE We thank the reviewer for this suggestion. However, we believe that our results are not only incremental, since we extend the role of NK cells in neuropathogenesis by studying a neurodegenerative disease in the CNS thereby showing effects that carry implication beyond those shown in DRG neurons after peripheral nerve injury. Nevertheless, we answered to the question on the role of lysosome exocytosis on MN death and the activation of NK cells performing in vitro assay of neurotoxicity in the presence of dynasore and wincostatin to block specific stages of lysosome exocytosis. We also performed this assay using NK cells isolated from perf KO mice. The new results demonstrated that specific stages of exocytosis are necessary for this effects, as well as the presence of perforin for the lytic machinery; these new data are now shown in Fig. 3d (Page 7 Line 15-22) and discussed (Page 13 Line 8).

7) *Is this also relevant to other neurodegenerative conditions such as Alzheimers and PD?*

RESPONSE This is an interesting point, that opens a wide discussion. To respect the journal constraints, however, and to focus on the results obtained in our experiments, we only insert a sentence in the discussion “It remains to be investigated whether a direct neurotoxic role of NK cells could be relevant to other neurodegenerative diseases” (end of Page 13).

8) *The study, whilst convincing, tends to rely on antibody mediated perturbations. I feel that orthogonal validation using e.g. genetic ablation studies / mice carrying a deletable mutant gene driven off a cell type-specific promoter would also add weight to findings*

RESPONSE The reviewer raises an important point. However, in the time frame indicated by the Editor (12 weeks) these experiments would be technically impossible. The time necessary to obtain mutant mice and to cross them with the SOD mutants before performing the experiments will require more than a year. The triple mutant mice could also introduce background strain variations that would require additional control experiments. Furthermore, the incremental information to be gained by such an experiment is uncertain. In particular, antibody-mediated blockade more nearly simulates the clinical situation. Replication of these results with a genetic model would not add information. Lack of replication could easily be ascribed to effects of mixed genetic background or germline absence of the targeted gene from fertilization. Therefore, we respectfully suggest that this experiment lies outside the scope of the present study.

9) *An additional marker beyond SMI32 should be used to identify motor neurons in the assays evaluating effects on NK depletion*

RESPONSE Following the reviewer suggestion, we also use ChAT to identify motor neurons, and the additional data are now shown in Suppl. fig. 3 d and inserted in the results (Page 8, Line 2).

10) *Can the authors include post-mortem delays for each of the cases*

We now include the post-mortem delay, in table 1.

11) *Can power calculations please be shown*

RESPONSE We add power calculation in all the figure legends.

12) *Can an ethics statement please be provided*

RESPONSE The ethics statement for animal experiments and human tissues is in the methods (Pages 15 and 16).

Reviewer #3 (Remarks to the Author):

Critique of “Natural killer cells modulate motor neuron-immune cell cross talk in Amyotrophic Lateral Sclerosis”

1. the figures are not numbered and the subfigures do not have heading, which makes the review difficult. Notwithstanding, the study is very carefully done and confirms nicely the hypothesis of role of NK cells in animal model using Ab-CCL2 and NK cell depletion.

RESPONSE We apologize for this lack of accuracy and thank the reviewer for the consideration.

2. The role of NK cells is important, but it needs to be demonstrated in ALS patients’ spinal cord by visualization of NK cells. The figure shows NK cells in cortex but there is no relation to neurons or vessels. A

description of localization is needed. Localization in anterior horns in vicinity of neurons could firm up the role of NK cells in ALS patients

RESPONSE To answer this important point, we performed immunofluorescence analysis to detect possible contacts between motor neurons (NeuN-Smi32) and NK cells (NKp46) / vessels (CD31) in spinal cord tissues from ALS patients. Results are shown in the new fig. 7 c,d and inserted in the text (Page 11, Lines 9-12).

3. The title should everywhere specify “ALS model” not “ALS”. This study points to the role of NK cells in animal models but has no neuropathological demonstration of NK cells in ALS patients’ spinal cord, although it has strong evidence in animal model

RESPONSE We thank the reviewer for this comment and agree that human data are of paramount importance in drawing conclusions about disease. We note that we included data for NK cells in patient motor cortex in the initial submission. Motivated by the reviewer’s request we now also show data for NK infiltration of ALS spinal cord (see previous comment and response).

4. This study has strong point of colocalization of Smi32 with Mult1 and other interactions, expression of IFN gamma with NK cells, etc .

RESPONSE: We thank the reviewer for these positive considerations.

5. The study overlooks the likely pathogenic role of cytotoxic CD8 T cells expressing granzyme and perforin in the spinal cord anterior horns of ALS patients (1, 2).

1. Fiala M, et al. (2010) IL-17A is increased in the serum and in spinal cord CD8 and mast cells of ALS patients. Journal of neuroinflammation 7(1):76-90.

2. Lam L, et al. (2016) Epigenetic changes in T-cell and monocyte signatures and production of neurotoxic cytokines in ALS patients. FASEB journal : official publication of the Federation of American Societies for Experimental Biology 30(10):3461-3473.

RESPONSE We thank the reviewer for raising this point. In our study, we cannot exclude a pathogenic role for CD8⁺ cells, but this has not been directly addressed in our experiments. However, we now discuss this point at page 13, last three lines.

6. This sentence is incomplete: Since NKG2D ligands are involved in triggering the cytotoxicity of NKG2D receptor-expressing NK cells

RESPONSE We apologize, now we completed the sentence.

7. Human data in Supl fig 1 just show ctrl vs. patients but there is no time of blood collection vs. onset of disease, which is critical to understand role in disease.

RESPONSE We thank the reviewer for this comment. This important information is now incorporated in TABLE 2 (months from diagnosis to blood sample) together with the deficit score.

Overall, this is a very strong work but it is difficult to review by non-aficionado of mouse models and mouse NK cells. It would be much easier to review if the legends were appended to pictures and each subfigure had a heading.

RESPONSE Again we apologize for the inconvenience and thank the reviewer for the patience in reading our manuscript.

Reviewer #4 (Remarks to the Author):

In the current manuscript the authors unravel a yet unknown role of NK cells in the onset of motor neuron degeneration in ALS and their impact on Treg recruitment and microglia. Combining research on human biomaterial with ALS mouse models the authors demonstrate that NK cells infiltrate the motor cortex and spinal cord most likely in a CCL2-dependent manner. NK-cell mediated cytotoxicity against motor neurons is driven by NKG2D expression. Accordingly, NK cell depletion reduced the pace of motor neuron degeneration, delayed the onset of motor impairment and increased survival in the mouse model. Finally, the authors also showed that IFN- γ produced by NK cells shifts microglia towards an inflammatory phenotype and reduces infiltration of Treg cells.

Although this study is well-designed, the following points need to be addressed.

Major points:

- The mechanism of NK-mediated motor neuron cytotoxic activity needs to be further elucidated. The authors should show, both in the human and mouse tissue, whether NK cells in contact with motor neurons express granzymes and/or perforin.*

RESPONSE We agree with the reviewer that it would be informative to see granzyme or perforin expression in NK cells in contact with motor neurons both in mouse and human tissues. To complete this analysis however we need more than the 12 weeks suggested by the Editor, so we are not ready to show the results of our experiments in this manuscript. However, we believe that other experiments performed to identify the interactions and the mechanism of NK cell-MN interactions both in human and mouse tissue (Fig.3 a,b,e; Fig 7a,b; suppl Fig 3a) and NK cell-mediated cytotoxic activity (Fig. 3d; suppl Fig 3b,c), which also requires perforin expression, could strongly support our experimental hypothesis.

Furthermore, the impact of granzyme and/or perforin blockade should be assessed in the mouse model.

RESPONSE Following the reviewer's suggestion, we performed in vitro cytotoxic assay with perf KO NK cells and hSOD1 motor neurons, as also requested at point 6 of review 2. Results are shown in Fig 3d and demonstrate that perforin and lysosome exocytosis are involved in the neurotoxic activity of NK cells against motor neurons.

- Although the authors demonstrated that NK cells isolated from the peripheral blood of sALS patients show no enhanced cytotoxic capacity towards K562 target cells when compared to healthy individuals, it would be interesting to see whether granzyme and perforin levels differ between these two groups.*

RESPONSE To answer this point, NK cells isolated from peripheral blood of ALS patients were analysed by FACS for granzyme and perforin expression. These experiments are now shown in Fig. 7 h,i. Data are described in the result section (page 11 last line).

- Are there any evidences, e.g. genetic, that NK cells are involved in the pathogenesis of ALS?*

RESPONSE As far as we know, there is no genetic evidence of NK cells being involved in ALS pathogenesis. There is one recent case report of aggressive lower motor neuron disease associated with NK/NKT cell leukaemia (La Bella Journal of Neurological Science 398, 2019). We decide not to mention this reference because we believe it is not relevant in this paper.

• *The authors describe diminished NK-cell proportions in the periphery of ALS patients. How do they explain this finding in the context that another study found increased NK cell numbers (reference 18 in the manuscript)?*

RESPONSE Indeed, the paper of Murdock et al., JAMA Neurol 74(12) describes an increased number of NK cells in patient's PB. We considered the relative amount of NK cells with respect to other populations of PBMC and demonstrated that the frequency of NK cells in PB of patients is reduced. Even if a direct relation among NK cells number and frequency is not obvious, further experiments should be performed in order to establish clear correlations with disease stage at the moment of blood collection, also considering ongoing pharmacological therapies. This is now discussed at page 12, lines 13-18.

• *What was the impact of CCL2 blocking on other immune subsets; particular CD8 T cells.*

RESPONSE We evaluated the effect of CCL2 Ab on CD3+ cells and the results are now inserted in the results (Page 5 Lines 13-16). We observed a reduction of CD3+ cells; however due to the low number of infiltrating CD3+ T cells after Ab-CCL2 treatment, we could not reliably quantify additional cell sub populations (such as CD8+ cells) in this pool.

• *Along this line, in addition to NK cells T cells can also produce IFN- γ . The authors should take this into consideration.*

RESPONSE We thank the reviewer for this criticism and we now add a comment to consider T cells as source of IFN- γ (page 13, last line).

Minor points:

• *For a broader readership the mouse models should be better explained in the results part.*

RESPONSE These descriptions are now inserted at page 5 (Lines 20-23).

• *In the first paragraph of page 8 the authors should already explain which ligand (Mult-1, Nectin-2, Rae-1, Pvr) belongs to which receptor (NKG2D, DNAM-1).*

RESPONSE We apologize for this lack of clarity, and now better explain the relationships ligands-receptors (Page 6, lines 17-18).

• *N-number for the human tissues should be already provided in the result part or respective figure legend.*

RESPONSE We inserted the number of human tissues in the figure legends.

REVIEWERS' COMMENTS:

Reviewer #1 (Remarks to the Author):

The authors have carefully addressed my points and revised the manuscript accordingly, especially expanding the number of ALS patients and controls to strengthen the relevance of NK cell findings to humans, and reconciling differences to a previous study depleting NK cells in mutant SOD1 mice.

Reviewer #2 (Remarks to the Author):

The authors have carefully answered my concerns either through new experiments or reasonable textual responses. In aggregate I am satisfied with these revisions and am supportive of publication

Reviewer #4 (Remarks to the Author):

The authors have addressed my comments satisfactory and the manuscript is now ready for acceptance.

REVIEWERS' LETTER

Reviewer #1 (Remarks to the Author):

The authors have carefully addressed my points and revised the manuscript accordingly, especially expanding the number of ALS patients and controls to strengthen the relevance of NK cell findings to humans, and reconciling differences to a previous study depleting NK cells in mutant SOD1 mice.

Reviewer #2 (Remarks to the Author):

The authors have carefully answered my concerns either through new experiments or reasonable textual responses. In aggregate I am satisfied with these revisions and am supportive of publication

Reviewer #4 (Remarks to the Author):

The authors have addressed my comments satisfactory and the manuscript is now ready for acceptance.

RESPONSE:

We thank all the reviewers for their precious work and for the positive evaluation of our revision process.